**SciPost Physics** **TT̄-deformed 1d Bose gas** Submission

Yunfeng Jiang[1,2],

**1** CERN Theory Department, 1 Esplanade des Particules, Geneva 23, CH-1211, Switzerland
**2** Shing-Tung Yau Center and School of Physics, Southeast University, Nanjing 210096, China
* jinagyf2008@gmail.com

April 19, 2021

## Abstract

TT̄ deformation was originally proposed as an irrelevant solvable deformation for 2d relativistic quantum field theories (QFTs). The same family of deformations can also be defined for integrable quantum spin chains which was first studied in the context of integrability in AdS/CFT. In this paper, we construct such deformations for yet another type of models, which describe a collection of particles moving in 1d and interacting in an integrable manner. The prototype of such models is the Lieb-Liniger model. This shows that such deformations can be defined for a very wide range of systems. We study the finite volume spectrum and thermodynamics of the TT̄-deformed Lieb-Liniger model. We find that for one sign of the deformation parameter ($\lambda < 0$), the deformed spectrum becomes complex when the volume of the system is smaller than certain critical value, signifying the break down of UV physics. For the other sign ($\lambda > 0$), there exists an upper bound for the temperature, similar to the Hagedorn behavior of the TT̄ deformed QFTs. Both behaviors can be attributed to the fact that TT̄ deformation changes the size the particles. We show that for $\lambda > 0$, the deformation increases the spaces between particles which effectively increases the volume of the system. For $\lambda < 0$, TT̄ deformation fattens point particles to finite size hard rods. This is similar to the observation that the action of TT̄-deformed free boson is the Nambu-Goto action, which describes bosonic strings — also an extended object with finite size.

# 1 Introduction

$T\overline{T}$ deformation [1,2], a special irrelevant deformation of relativistic quantum field theories (QFTs) has been studied intensely in recent years (for a pedagogical review see [3]). The deformed theory exhibits many novel features compared to usual local QFTs. At the same time, solvability of the deformation allows one to perform analytical studies. In this sense, the study of $T\overline{T}$ deformation has deepened our understanding of QFTs in a well controlled set-up.

It is natural to ask whether similar deformations can be defined for other types of models, such as non-relativistic quantum many-body systems and lattice models like spin chains. This question is interesting for several reasons. Firstly, QFTs describes the low energy, long wave length

behavior of the underlying quantum many-body systems. The study of solvable deformations of the underlying system can potentially shed new lights on $T\overline{T}$ deformation of the emergent QFT. Secondly, solvable deformations of many-body systems are interesting in their own right, especially when the undeformed theories are integrable. This will lead to new kinds of integrable models that are worth studying. Finally, by investigating other types of models with the same feature as the $T\overline{T}$ deformed relativistic QFTs, we can gain a more universal picture about the deformation. We can try to understand some remaining puzzles of $T\overline{T}$ deformation in a simpler set-up. For example, it is by far well-known that for one sign of the deformation parameter, the deformed spectrum becomes complex at certain point. How to understand this behavior ? This might be a hard question for QFTs since finding finite volume spectrum for QFTs is by itself a challenging question. Suppose we can $T\overline{T}$ deform a quantum-many body system whose spectrum is well-understood, it would be easier to answer this question in the simpler case.

A perfect playground for such generalizations is integrable models. The main reason is that in many cases, different types of integrable models can be studied in a universal way using the same technique. For example, the spectrum of the following three different types of models: XXZ spin chain (lattice model), Lieb-Liniger model (non-relativistic continuous many-body system) and the Sinh-Gordon model[1] (relativistic QFT) can all be studied by Bethe ansatz ! The constructions of the eigenstates are basically the same. The only differences are the dispersion relations for the excitations and their factorized two-body S-matrices. In addition, $T\overline{T}$ deformation for integrable QFTs is particularly simple. The deformation preserves integrability and only changes the factorized S-matrix by a phase factor [1,4] called CDD factor. The seemingly harmless CDD factors in fact violate the polynomial boundedness of the S-matrix for local QFTs, and is responsible for all kinds of unusual behaviors of the deformed theory in the UV. Therefore, from integrability point of view, it is clear that we shall look for integrable deformations which changes the S-matrix by a similar phase factor.

It is intriguing that such deformations for quantum spin chains have been studied a decade ago in disguise from a rather different motivation. In the study of integrability in AdS/CFT correspondence, it is known that the dilatation operator of planar $\mathcal{N} = 4$ SYM theory is described by a long-range interacting spin chain [5,6]. This spin chain is integrable, but is unusual because the interacting range grows order by order in 'tHooft coupling in perturbation theory. In an attempt to understand this new kinds of spin chains in a more systematic manner, the authors of [7,8] classified and studied integrable long-range deformations for quantum spin chains. One class of the deformations, dubbed *bilocal deformation* is exactly what we are seeking for. This deformation preserves integrability and deforms the S-matrix by a simple phase factor. The connection between the bilocal deformation and $T\overline{T}$-*like* deformations was only pointed out recently in [9] (see also [10]), where the authors also identified the deformation operators and proved the factorization property of the mean value of the deformation operators.

One important subtlety for spin chains is that, due to their discrete nature, strictly speaking it is not possible to construct the $T\overline{T}$ deformation. The reason is that the construction of [7,8] relies on the fact that the conserved charges have a *local form*, namely can be written as a sum/integral over *densities*. This requirement is met for all the conserved charges of the spin chain, except for the momentum operator which is needed to construct the $T\overline{T}$ deformation. Therefore we cannot construct the 'real' $T\overline{T}$ operator, but rather all its cousins constructed from higher conserved charges.

In order to construct the true $T\overline{T}$ deformation, we need to consider quantum many-body systems which live on a *continuous* space. This is the goal of the current work. We consider the models

---

[1]We consider the spectrum in the large volume limit where finite size corrections can be neglected.

which describes a collection of particles moving in 1d and interacting in an integrable manner. The prototype of such models is the Lieb-Liniger model [11], or the non-linear Schrödinger model which describes 1d Bose gas with $\delta$-function interaction. This model has been well studied in integrability since 1960s'. In recent years, it has received considerable renewed interested in the study of out-of-equilibrium statistical physics. This model can be engineered by cold atom experiments, and at the same time can be studied analytically thanks to integrability.

The main results of this paper are summarized as follows. We construct a family of integrable bilinear deformations from conserved currents of the model. We show that the deformations modify the S-matrix by a CDD-like phase factor. The $T\overline{T}$ deformation belongs to this family and is the 'next-to-simplest' bilinear deformation. The 'simplest' bilinear deformation, constructed from the conserved currents of particle number operator and momentum, is also studied and turns out to be interesting. We show that this deformation effectively changes the size of the point-like particle. For one sign of the deformation parameter ($\lambda < 0$ in our convention), the deformation fattens the point-particle to *finite size* hardcore particles, or hard rods whose size is given by $|\lambda|$. For $\lambda > 0$, the hard rod has a 'negative length' $\lambda$, or equivalently the space between the particles is increase by $\lambda$.

The $T\overline{T}$ deformation exhibit the same feature. For $\lambda < 0$, the deformation turns point-like particles into finite size rods, where the size of the rod is given by its deformed energy, which needs to be determined self-consistently. For $\lambda > 0$, the deformation effectively increases the volume of the system. This reminds us a well-known fact of the $T\overline{T}$ deformation for relativistic QFTs. It is shown that the deformed Lagrangian of the free boson is the Nambu-Goto action in the static gauge [2,12,13], which describes bosonic strings. This implies to certain extent that the deformation fattens particles to strings — another extended object with finite size. This intuition turns out to be very useful for understanding the main features of the deformed spectrum and thermodynamics, which constitute the other two main results of the paper.

We find the deformed finite volume spectrum both for finite particle states and in the continuum limit. We find that for $\lambda < 0$, there exist a critical value $\lambda_c$ such that for $\lambda < \lambda_c$ the spectrum become complex. This is the same behavior as the spectrum in $T\overline{T}$ deformed CFTs. For $\lambda > 0$, the deformed spectrum is always well-defined and approaches to 0 as $\lambda \to +\infty$. This behavior can be explained using the hard rod intuition. For fixed system size and particle number, in the $\lambda < 0$, the size of each hard rod has to be bounded. Whenever this bound is violated, we find complex spectrum. Alternatively, if we fix the deformation parameter $\lambda < 0$ and number of particles $N$. We find a lower bound $R_c$ for the volume of the system such that when $R < R_c$ the spectrum becomes complex. This signifies the break down of UV physics.

We study the thermodynamics of the deformed theory by the method of thermodynamic Bethe ansatz (TBA) [14]. We find that the effect of the deformation is shifting the chemical potential by an amount $A_\lambda(\beta, \mu)$ which depends on the temperature $1/\beta$, the undeformed chemical potential $\mu$ and the deformation parameter $\lambda$. The quantity $A_\lambda(\beta, \mu)$ is the solution of a self-consistency relation. For $\lambda < 0$ and fixed $\beta, \mu > 0$, real solution of $A_\lambda(\beta, \mu)$ always exists. For $\lambda > 0$ and fixed $\beta, \mu > 0$, there is a critical value $\tilde{\lambda}_c(\beta, \mu)$ such that for $\lambda > \tilde{\lambda}_c(\beta, \mu)$ real solution does not exist, which signifies a singularity. Alternatively, we can fix $\lambda, \mu > 0$ and vary $\beta$. The solution for the self-consistency relation for $A_\lambda(\beta, \mu)$ only exists for $\beta \geq \beta_H(\mu, \lambda)$ for certain critical value $\beta_H(\mu, \lambda)$. This implies an upper bound for the temperature $T_H = 1/\beta_H$, which is the non-relativistic version of the Hagedorn temperature. Again, this is the same behavior as one finds in the $T\overline{T}$ deformed QFTs [15–17]. The reason for the existence of the Hagedorn temperature can also be explained by the hard rod intuition.

We want to stress that, although for concreteness we choose to work with the Lieb-Liniger model.

Many of our results can be generalized to other integrable models without much difficulty. At least at the level of S-matrix and integrability[2], the generalizations to similar models such as Toda chain and the Calogero-Sutherland model [19] are straightforward. Therefore we believe the behaviors we found in this paper are generic for the $T\overline{T}$ deformed theories. Finally, we would also like to mention that there are other proposals for the $T\overline{T}$ deformation of quantum mechanical systems which are motivated from holography [20, 21]. These are also highly interesting deformations that deserve further investigation, but they are different from the bilinear deformations that we are studying in the current paper.

The rest of this paper is organized as follows. We give a brief review of Lieb-Liniger model to set-up the stage and fix notations in section 2. Then we construct the family of integrable bilinear deformations in section 3. The study of finite volume spectrum is performed in section 5,6 and 7. Thermodynamics is considered in section 8. We make further comments on bilinear deformation and generalized TBA and Gibbs ensemble in section 9. We conclude in section 10. Several appendices are devoted to explaining technical details and giving useful backgrounds for the discussions in the main text.

**Note Added** As I am writing up the current paper, [22] appeared on arXiv which has partial overlap with our results. In particular, we arrive at the hard rod interpretation of the $T\overline{T}$ deformation independently. The results of the two papers are consistent, although the emphasis and approaches are different.

## 2 The Lieb-Liniger model

In this section, we introduce the Lieb-Liniger model. For more detailed discussions, we refer to [23, 24]. In the second quantized form, it can be described by a non-relativistic quantum field theory of a scalar field $\Phi(x, t)$ with the Lagrangian density

$$\mathcal{L} = \frac{i}{2} \left( \Phi^\dagger \partial_t \Phi - \partial_t \Phi^\dagger \Phi \right) - \partial_x \Phi^\dagger \partial_x \Phi - c\, \Phi^\dagger \Phi^\dagger \Phi \Phi. \tag{1}$$

where $c$ is the coupling constant. In this paper, we take $c > 0$ which corresponds to the repulsive interaction. The scalar field satisfies the usual equal time commutation relations

$$[\Phi(x, t), \Phi(y, t)] = 0, \qquad [\Phi^\dagger(x, t), \Phi^\dagger(y, t)] = 0, \qquad [\Phi(x, t), \Phi^\dagger(y, t)] = \delta(x - y). \tag{2}$$

The Hamiltonian is given by

$$H = \int \mathrm{d}x \left[ \partial_x \Phi^\dagger(x) \partial_x \Phi(x) + c\, \Phi^\dagger(x) \Phi^\dagger(x) \Phi(x) \Phi(x) \right] \tag{3}$$

where the integration domain can be non-compact or compact with length $R$. We will specify to each case later. It is also useful to define the particle number operator and the momentum operator

$$\hat{N} = \int \Phi^\dagger(x) \Phi(x) \mathrm{d}x, \qquad P = -\frac{i}{2} \int \left[ \Phi^\dagger(x) \partial_x \Phi(x) - \partial_x \Phi^\dagger(x) \Phi(x) \right] \tag{4}$$

The Hilbert space is decomposed into multi-particle sectors. The Fock vacuum is defined by

$$\Phi(x)|0\rangle = 0, \qquad x \in \mathbb{R}. \tag{5}$$

---

[2]Since not all such quantum mechanical systems have second quantized description in terms of local non-relativistic QFTs, we expect there might be some subtleties for constructing the bilinear operators explicitly using fundamental fields. However, a similar CDD deformation for the factorized S-matrix [1, 18] can be defined easily.

The $N$-particle sector is spanned by states like

$$|\mathbf{x}\rangle = \Phi^\dagger(x_1)\ldots\Phi^\dagger(x_N)|0\rangle, \tag{6}$$

In this sector, the Hamiltonian is given by the more familiar form in quantum mechanics

$$H = -\sum_{i=1}^N \frac{\partial^2}{\partial x_i^2} + 2c\sum_{i<j}\delta(x_i - x_j) \tag{7}$$

where $x_i$ are the positions of the particles. The momentum operator reads

$$P = -i\sum_{j=1}^N \frac{\partial}{\partial x_i}. \tag{8}$$

**Integrability** The Lieb-Liniger model is integrable and can be solved by Bethe ansatz. The eigenstate is constructed by Bethe ansatz

$$|\mathbf{u}_N\rangle = \frac{1}{\sqrt{N!}}\int \mathrm{d}^N x\,\psi_N(\mathbf{u}|\mathbf{x})\,|\mathbf{x}\rangle. \tag{9}$$

where the position space wave function reads

$$\psi_N(\mathbf{u}|\mathbf{x}) = \frac{1}{\sqrt{N!}}\sum_{\sigma\in S_N}\exp\left[i\sum_{j=1}^N x_j u_{\sigma_j}\right]\prod_{j>k}\frac{u_{\sigma_j} - u_{\sigma_k} - ic\,\mathrm{sgn}(x_j - x_k)}{u_{\sigma_j} - u_{\sigma_k}}. \tag{10}$$

Here $\mathrm{sgn}(x)$ is the sign function and the $N$ parameters $\mathbf{u} = \{u_1,\ldots,u_N\}$ are called rapidities. Imposing periodic boundary condition leads to the quantization condition of the rapidities

$$e^{iu_j R}\prod_{k\neq j}^N \frac{u_j - u_k - ic}{u_j - u_k + ic} = 1. \tag{11}$$

From here, we extract the S-matrix of the particles

$$S(u,v) = \frac{u - v - ic}{u - v + ic}. \tag{12}$$

The eigenvalues of energy and momentum are given in terms of the rapidities as

$$E_N(\mathbf{u}) = \sum_{j=1}^N u_j^2, \qquad P_N(\mathbf{u}) = \sum_{j=1}^N u_j. \tag{13}$$

The norm of the wave function is given by

$$\mathcal{N}_N = \int |\psi_N|^2 \mathrm{d}^N x = \prod_{j<k}^N \frac{(u_j - u_k)^2 + c^2}{(u_j - u_k)^2}\times\det G \tag{14}$$

where $G$ is the Gaudin matrix whose matrix elements are given by

$$G_{jk} = \delta_{j,k}\left(R + \sum_{l=1}^N \varphi(u_j,u_l)\right) - \varphi(u_j,u_k) \tag{15}$$

with

$$\varphi(u,v) = -i\frac{\partial}{\partial u}\log S(u,v) = \frac{2c}{(u-v)^2 + c^2}. \tag{16}$$

The function $\varphi(u,v)$ is called the TBA kernel and plays an important role in our calculations below.

**Simple limits**    There are two limits of the Lieb-Liniger model which are particularly simple and are very useful for analytical studies below. In the limit $c \to 0$, the interaction is turned off and we have a system of free bosons. In the limit $c \to \infty$, the repulsion between the particles are so strong that they behave like free fermions. Therefore we will call the two limits by free boson and free fermion limits in this paper. Sometimes the free fermion limit is also called the Girardeau-Tonks limit. The TBA kernel (16) simplifies in both limits. In the free fermion limit we have $\varphi(u, v) = 0$ while in the free boson limit we have $\varphi(u, v) = 2\pi\delta(u - v)$.

# 3   Integrable bilinear deformations

In this section, we define a family of integrable bilinear deformations for the Lieb-Liniger model. The $T\overline{T}$ deformation is a member of this family. Our construction here is a natural extension of the spin chain case [7–9].

## 3.1   Bilinear and bilocal deformations

Consider a conserved current $\partial_a \mathcal{J}^a(x, t) = 0$. The two components $\mathcal{J}^a = (q, j_Q)$ are the charge and current densities, using which the conservation equation can be written as

$$\partial_t q(x, t) + \partial_x j_Q(x, t) = 0 \tag{17}$$

The spacial integral of $q(x, t)$ gives the corresponding conserved charge

$$Q = \int \mathrm{d}x \, q(x, t), \qquad \frac{\mathrm{d}}{\mathrm{d}t} Q = 0. \tag{18}$$

Let us now consider two conserved currents $\mathcal{J}_1^a(x, t)$ and $\mathcal{J}_2^a(x, t)$. We can construct the following composite operator

$$\mathcal{O}_{\mathcal{J}\mathcal{J}}(x, t) = \varepsilon^{ab} \mathcal{J}_1^a \mathcal{J}_2^b(x, t). \tag{19}$$

In terms of components,

$$\mathcal{O}_{\mathcal{J}\mathcal{J}} = q_1 \, j_{Q_2} - q_2 \, j_{Q_1}. \tag{20}$$

The $T\overline{T}$ operator corresponds to taking $Q_1 = P$ and $Q_2 = H$ where $P$ and $H$ are the momentum and the Hamiltonian of the system. The bilinear deformation is defined in the Hamiltonian formalism as

$$\frac{\mathrm{d}}{\mathrm{d}\lambda} H_\lambda = \int \mathrm{d}x \, \mathcal{O}_{\mathcal{J}\mathcal{J}}^{(\lambda)}(x, t) \tag{21}$$

where $\lambda$ is the deformation parameter and $H_\lambda$ is the deformed Hamiltonian.

Let us comment on the construction of the bilinear operator. Since in a wide class of systems we can define the momentum and Hamiltonian, it seems that we can define the $T\overline{T}$ for all these systems. However, the crucial point here is that they need to be written in a *local* form, *i.e.* they should take the form of integrals or sums over *densities*. This requirement, however, is not always met. For example, in quantum spin chains, the momentum of the system is defined as logarithm of the shift operator, which is a non-local operator and cannot be written in a local form. This is of course related to the discrete nature of the spin chain. Therefore strictly speaking, we cannot

define T$\overline{\text{T}}$ deformation for quantum spin chains. Nevertheless, for integrable spin chains, there are higher conserved charges which can be written in local forms, we can define a family of bilinear deformations using the higher conserved charges.

For the Lieb-Liniger model, since the space is continuous, the momentum operator can be written in a local form and T$\overline{\text{T}}$ deformation can be defined. Let us denote the conserved charges by $\{Q_0, Q_1, Q_2, \ldots\}$ where $Q_0 = \hat{N}$ is the number operator, $Q_1 = P$ is the momentum and $Q_2 = H$ is the Hamiltonian. The rest are the higher conserved charges due to integrability of the model. We denote the bilinear operator (20) corresponding to the charges $Q_a, Q_b$ by $O_{a,b}$. The T$\overline{\text{T}}$ operator corresponds to $O_{1,2}$ in this notation.

**Bilinear and bilocal deformations**   As mentioned before, the bilinear deformation is tightly related to the bilocal operator that was first defined for quantum spin chains [7, 8]. We briefly review this connection here because it is useful for later discussions. The bilocal operator is defined as

$$\mathcal{X}_{\mathcal{J}\mathcal{J}} = \int_{x<y} \mathrm{d}x\mathrm{d}y \, q_1(x) q_2(y). \tag{22}$$

where $q_1(x)$ and $q_2(x)$ are two charge densities. We consider the integral on a finite ring of length $R$. The integral can be written in two equivalent ways

$$\int_{x<y} \mathrm{d}x\mathrm{d}yF(x,y) = \lim_{\varepsilon \to 0} \int_0^R \mathrm{d}y \int_0^{y-\varepsilon} \mathrm{d}xF(x,y) = \lim_{\varepsilon \to 0} \int_0^R \mathrm{d}x \int_{x+\varepsilon}^R \mathrm{d}yF(x,y). \tag{23}$$

Using the Schrödinger equation and the conservation equation

$$\partial_t q_a(x) = i[q_a(x), H], \qquad i[q_a(x), H] = -\partial_x j_{Q_a} \tag{24}$$

We can show that

$$i[\mathcal{X}_{\mathcal{J}\mathcal{J}}, H] = \int_0^R \mathcal{O}_{\mathcal{J}\mathcal{J}}(x,t)\mathrm{d}x - [Q_1 \, j_{Q_2}(0) - Q_2 \, j_{Q_1}(0)] \tag{25}$$

where we have used the periodic boundary condition $j_{Q_a}(R) = j_{Q_a}(0)$. Denoting the operator

$$\mathcal{Y}_{\mathcal{J}\mathcal{J}} = Q_2 \, j_{Q_1}(0) - Q_1 \, j_{Q_2}(0). \tag{26}$$

We find

$$\frac{\mathrm{d}}{\mathrm{d}\lambda}H_\lambda = \int_0^R O_{\mathcal{J}\mathcal{J}}(x)\mathrm{d}x = i[X_{\mathcal{J}\mathcal{J}}, H_\lambda] + \mathcal{Y}_{\mathcal{J}\mathcal{J}}. \tag{27}$$

The merit of writing the bilinear deformation in this form is that it separates the effects in infinite volume and finite volume effects. In the limit $R \to \infty$, the term $\mathcal{Y}_{\mathcal{J}\mathcal{J}}$ can be neglected, and we are left with the first term. This is precisely the bilocal deformation. For integrable models, we denote the bilocal operator $\mathcal{X}_{\mathcal{J}\mathcal{J}}$ constructed from charges $Q_a, Q_b$ by $X_{a,b}$.

**Infinite volume**   In the infinite volume, the bilinear deformation is identical to the bilocal deformation

$$\frac{\mathrm{d}}{\mathrm{d}\lambda}H_\lambda = i[\mathcal{X}_{\mathcal{J}\mathcal{J}}, H_\lambda] \tag{28}$$

This equation can be solved formally by

$$H_\lambda = U_\lambda H_0 U_\lambda^{-1}, \qquad U_\lambda = \mathcal{P} \exp\left[-i \int_0^\lambda \mathcal{X}_{\mathcal{J}\mathcal{J}}^{(\lambda')} \, \mathrm{d}\lambda'\right] \tag{29}$$

where $\mathcal{P}$ denotes path ordering for the operator exponential. The spectrum is undeformed in infinite volume. The eigenstates deform in a simple way. Consider an undeformed eigenstate $|\phi_n\rangle$ with eigenvalue $E_n$ such that

$$H_0|\phi_n\rangle = E_n|\phi_n\rangle. \tag{30}$$

The deformed eigenstate is given by

$$|\phi_n\rangle_\lambda = U_\lambda|\phi_n\rangle. \tag{31}$$

and we have

$$H_\lambda|\phi_n\rangle_\lambda = U_\lambda H_0 U_\lambda^{-1} U_\lambda|\phi_n\rangle = E_n|\phi_n\rangle_\lambda. \tag{32}$$

**Finite volume**  To have deformed spectrum, we need to consider the bilinear deformations in the finite volume. The deformation of the energy comes from the term $\mathcal{Y}_{\mathcal{J}\mathcal{J}}$. By putting (27) in the mean value, one can write down the flow equation for the energy spectrum. This has been done for the $T\overline{T}$ deformation for relativistic QFTs in [25], which leads to the same flow equation as derived from the factorization formula [1,2]. However, for generic non-relativistic theories, the expectation value of the current density operators are not known in a closed form. This makes it hard to find the deformed energy by solving the flow equation.

On the other hand, the situation is much better for integrable models. There are two ways to see this. The first way is to consider the flow equation in finite volume. For integrable models, the mean values of the current operators in most cases are known and can be written down explicitly in terms of Bethe roots. This additional information allows us to solve the flow equation and find the deformed spectrum. Alternatively, we can first consider the deformation in infinite volume. Although the spectrum is not deformed in this case, but one can determine the deformed scattering data which includes the dispersion relation of the excitations and their factorized S-matrix in this limit. It turns out that the dispersion relation is not modified by the bilocal deformation. The S-matrix is deformed in a simple way by multiplying a CDD-like phase factor. Once the deformed S-matrix is known, we can go back to the finite volume case by imposing periodic boundary condition. The key point is that all the finite volume effects are taken into account by the quantization condition of the momenta of the excitations, which is the deformed Bethe equations. From the deformed Bethe equations, we can also derive the same flow equation for the spectrum. We will discuss this method in detail in the next section.

Before ending this section, let us comment on the integrability of the deformed theory. In the infinite volume limit, the bilinear deformation preserves integrability. This is because it is equivalent to the bilocal deformation, which is an algebra preserving deformation [7]. Consider a set of charges $Q_a$, $a \in \mathbb{N}$ satisfying the following commutation relations

$$[Q_a, Q_b] = f_{abc}Q_c \tag{33}$$

for some structure constant $f_{abc}$. We deform these charges by

$$\frac{\mathrm{d}}{\mathrm{d}\lambda}Q_a(\lambda) = i[\mathcal{X}_{\mathcal{J}\mathcal{J}}, Q_a(\lambda)]. \tag{34}$$

where $\mathcal{X}_{\mathcal{JJ}}$ is the bilocal operator. It is then easy to prove

$$\frac{\mathrm{d}}{\mathrm{d}\lambda}[Q_a(\lambda), Q_b(\lambda)] = i[\mathcal{X}_{\mathcal{JJ}}, [Q_a(\lambda), Q_b(\lambda)]] \tag{35}$$

by Jacobi identity. Suppose the deformed commutation relation reads

$$[Q_a(\lambda), Q_b(\lambda)] = f_{abc}(\lambda) Q_c(\lambda) \tag{36}$$

where $f_{abc}(\lambda)$ is the deformed structure constant. Then (35) implies that

$$\frac{\mathrm{d}}{\mathrm{d}\lambda} f_{abc}(\lambda) = 0. \tag{37}$$

Namely, the structure constant is not deformed. So the deformed charges satisfy the same algebra. As a special case, if we start with a set of commuting charges, they will remain commuting after the bilocal deformation. This shows that the deformation preserves integrability in the infinite volume. In the finite volume, as we discussed before, the finite volume effects are captured by the quantization condition. So integrability should again be preserved. Therefore, we can view the deformed theory as a new integrable model with the deformed S-matrix.

## 3.2   Deformed S-matrix

Now we consider the effect of the bilocal deformation on the scattering data. For Bethe ansatz solvable integrable models, there are two main ingredients in writing down the wave function, which are dispersion relation of the excitations and their two-body S-matrix. These quantities are derived in the *infinite volume* by considering the one- and two-particle states, respectively. Our derivation below is a straightforward generalization of [7] to the continuous model.

It is easy to see that the one-particle dispersion relation is not modified by the bilocal deformation. Let us consider the modification of the S-matrix. Without loss of generality, we consider the case where the two excitations are located at the positions $x_1$ and $x_2$ with $x_1 < x_2$. We denote the two rapidities by $u$ and $v$. The two particle state is given by

$$|u, v\rangle = A(u, v)|u < v\rangle + A(v, u)|v < u\rangle. \tag{38}$$

The partially ordered state is defined by

$$|u < v\rangle = \int_{x_1 < x_2} e^{ip(u)x_1 + ip(v)x_2} |x_1, x_2\rangle \tag{39}$$

The state $|v < u\rangle$ is defined by swapping $u$ and $v$. The S-matrix is given by the ratio of the coefficients

$$S(u, v) = \frac{A(v, u)}{A(u, v)}. \tag{40}$$

Now we derive the deformed S-matrix. The two-particle state is an eigenstate of the undeformed Hamiltonian. From (31), the deformed state is

$$|u, v\rangle_\lambda = U_\lambda |u, v\rangle = A(u, v) U_\lambda |u < v\rangle + A(v, u) U_\lambda |v < u\rangle. \tag{41}$$

Let us denote $|u < v\rangle_\lambda \equiv U_\lambda |u < v\rangle$. Take the variation with respect to $\lambda$, we find

$$\frac{\mathrm{d}}{\mathrm{d}\lambda}|u < v\rangle_\lambda = -iX_{\mathcal{JJ}}^{(\lambda)}|u < v\rangle_\lambda. \tag{42}$$

Partially ordered two particle state is an eigenstate of the bilocal operator $X_{\mathcal{J}\mathcal{J}}^{(\lambda)}$ (see appendix B for a derivation)

$$X_{\mathcal{J}\mathcal{J}}^{(\lambda)}|u < v\rangle_\lambda = \big(h_1(u)h_2(v) + f_{12}(u) + f_{12}(v)\big)|u < v\rangle_\lambda \tag{43}$$

where $f_{12}(u)$ is the eigenvalue where both operators $q_1(x)$ and $q_2(x)$ act on the same particle. The explicit form of $f_{12}(u)$ is not important for deriving the S-matrix. The eigenvalues on the right hand side of (43) is independent of $\lambda$, we can integrate both sides easily, which leads to

$$|u < v\rangle_\lambda = \exp\big[-i\lambda\big(h_1(u)h_2(v) + f_{12}(u) + f_{12}(v)\big)\big]|u < v\rangle \tag{44}$$

Swapping $u$ and $v$, we obtain a similar expression

$$|v < u\rangle_\lambda = \exp\big[-i\lambda\big(h_1(v)h_2(u) + f_{12}(u) + f_{12}(v)\big)\big]|v < u\rangle \tag{45}$$

Taking the ratio of the deformed partial ordered states, we obtain the deformed S-matrix

$$S_\lambda(u,v) = e^{-i\lambda\big(h_1(u)h_2(v) - h_2(u)h_1(v)\big)}S(u,v). \tag{46}$$

We find that the deformed S-matrix is simply related to the undeformed S-matrix by multiplying a phase factor, which is similar to the CDD factor in the relativistic case. In fact, if we take the relativistic dispersion relations for the excitations, we obtain precisely the CDD factors.

The factorized S-matrix is the central quantity of integrable models, which contains most (if not all) of the dynamical information of the model. Because the deformed model is still integrable, knowing the deformed S-matrix allows us to study the deformed theory using the standard toolkit of integrability. This will be demonstrated by the study of deformed spectrum and thermodynamics in the following sections.

## 4 $O_{0,1}$ deformation and the hard rod gas

Before discussing the $T\overline{T}$ deformation of the Lieb-Liniger model, we first consider the simplest the bilinear deformation, which is triggered by the $O_{0,1}$ operator. The conserved charges correspond to this operator are the particle number operator $Q_0 = \hat{N}$ and the momentum $Q_1 = P$. This simpler deformation is interesting in its own right and will offer us important intuition about the $T\overline{T}$ deformation. Let us denote the charge and current density of $Q_0$ by $\eta(x)$ and $j_{\hat{N}}(x)$. The bilinear operator then reads

$$O_{\mathcal{J}\mathcal{J}}(x) = \eta(x)j_P(x) - p(x)j_{\hat{N}}(x) \tag{47}$$

The corresponding bilocal operator reads

$$\mathcal{X}_{\mathcal{J}\mathcal{J}} = \int_{x<y} \mathrm{d}x\mathrm{d}y\,\eta(x)p(y). \tag{48}$$

Let us denote the $N$-particle eigenstate by $|\mathbf{u}_N\rangle$. The mean value of the of the charge densities are

$$\langle\mathbf{u}_N|\eta(x)|\mathbf{u}_N\rangle = \frac{N}{R}, \qquad \langle\mathbf{u}_N|p(x)|\mathbf{u}_N\rangle = \frac{P_N(R)}{R}. \tag{49}$$

where $R$ is the volume of the system. The mean value of the momentum current operator is

$$\langle \mathbf{u}_N | j_P(x) | \mathbf{u}_N \rangle = \frac{\partial}{\partial R} E_N(R, \lambda). \tag{50}$$

The mean value of $j_{\hat{N}}(x)$ is not important for the discussion in this section, a more general formula for such quantities will be discussed in section 5. Using the Feynman-Hellerman theorem

$$\langle \mathbf{u}_N | \partial_\lambda H_\lambda | \mathbf{u}_N \rangle = \partial_\lambda E_N(R, \lambda) \tag{51}$$

and the definition (27), we can write down the flow equation for the spectrum

$$\partial_\lambda E_N(R, \lambda) = N \, \partial_R E_N(R, \lambda) - \langle \mathbf{u}_N | j_{\hat{N}} | \mathbf{u}_N \rangle P_N(R). \tag{52}$$

The eigenvalue of the charges are

$$Q_0 | \mathbf{u}_N \rangle = \sum_{k=1}^N h_0(u_k) | \mathbf{u}_N \rangle, \qquad Q_1 | \mathbf{u} \rangle = \sum_{k=1}^N p(u_k) | \mathbf{u}_N \rangle \tag{53}$$

where $h_0(u) = 1$ and $p(u) = u$. From (46), the S-matrix is deformed as

$$S(u, v) \to e^{i\lambda(p(u) - p(v))} S(u, v) \tag{54}$$

Let us consider the zero momentum sector for simplicity. In this case, the flow equation simplifies to $\partial_\lambda E_n(R, \lambda) = N \partial_R E(R, \lambda)$, which implies that the deformation is simply changing the length of the system. It is also easy to see this form the deformed BAE, which in the zero momentum sector reads

$$p(u_j)(R + \lambda N) + \sum_{k \neq j}^N \theta(u_j, u_k) = 2\pi I_j, \qquad j = 1, \cdots, N. \tag{55}$$

We see that in the zero momentum sector, this deformation changes the size of the system by $\lambda N$. From this we can immediately deduce some qualitative behavior of the deformed model. We expect the physics for different sign of $\lambda$ to be different. For $\lambda > 0$, we can take $\lambda$ to be any positive value. In particular, we can take $\lambda \to +\infty$ limit. In this limit, the length tends to infinity and the particles are so far away from each other that they seldom interact. So we obtain an almost free theory for any $\theta(u, v)$. On the other hand, for $\lambda < 0$, since physically we shall require $R + \lambda N \geq 0$ we have $\lambda \geq -\frac{R}{N}$. Namely, for fixed $N$ and $R$, there's a critical value $\lambda_c = -N/R$ beyond which the system breaks down. The break down of the system can be seen in various physical quantities. For example, taking $\theta(u, v) = 0$ in the free fermion limit, we find that the momentum and energy are divergent at the critical value.

There is an alternative interpretation of our observation, which is related to the so-called hard rod model. This is the model describes a free system of hard rods with *finite size*. The Hamiltonian of the hard rod model is given by

$$H = -\sum_{j=1}^N \frac{\partial^2}{\partial x_j^2} + \sum_{i<j}^N v(x_i - x_j) \tag{56}$$

with the interaction

$$v(x) = \begin{cases} \infty, & \text{for } |x| < a \\ 0, & \text{for } |x| > a \end{cases} \tag{57}$$

where $a > 0$ is a positive number describing the size of the hard rod. This is an integrable model with the phase shift [26, 27]

$$\theta_{\text{HR}}(u, v) = -i \log S_{\text{HR}}(u, v) = -\pi \text{sgn}(u - v) - a(u - v) \tag{58}$$

Now we take the S-matrix of the Lieb-Liniger model in the free boson limit $c \to 0$. The deformed phase shift (47) is

$$\lim_{c \to 0} \theta(u, v) + \lambda(p(u) - p(v)) = -\pi \text{sgn}(u - v) + \lambda(u - v). \tag{59}$$

We find that for $\lambda < 0$, the S-matrix for the deformed free boson is precisely the hard rod model ! Therefore, we find that the deformation for $\lambda < 0$ can be interpreted as fattening a point-like particle to a finite size hard rod of length $|\lambda|$, see figure 1. It is then obvious that this value has to

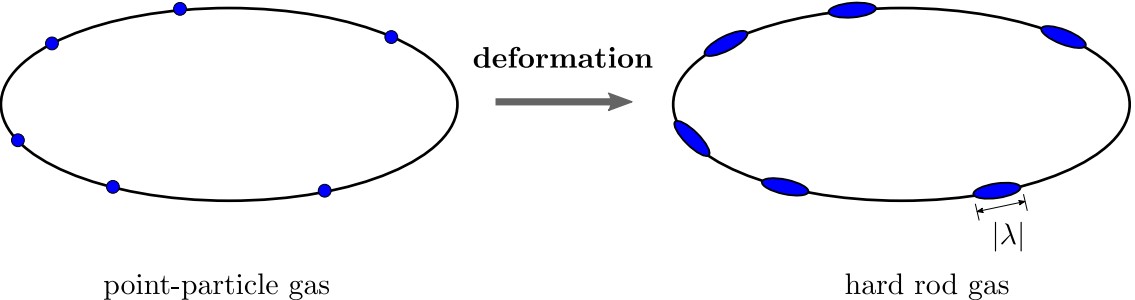

$$\text{point-particle gas} \qquad\qquad \text{hard rod gas}$$

Figure 1: The simple bilinear deformation turns a free bose gas into a free hard rod gas.

be bounded for fixed $N$ and $R$. Since each rod has the length $|\lambda|$. In order to fit $N$ such rods in a length $R$ ring, we must have $|\lambda|N \leq R$.

A few remarks are in order. Firstly, the 'fattening' point-like particles to finite size rods is very similar to what happens in the relativistic quantum field theory case. There by deforming the action of the free boson, one finds the classical Nambu-Goto action, which describes bosonic strings. In that context, the deformation parameter plays the role of the string tension while here it corresponds to the length of the hard rod.

Secondly, the qualitative picture for the $T\overline{T}$ deformation of the Lieb-Liniger model is similar to the $O_{0,1}$ deformation. In the zero momentum sector, we simply need to replace the particle number $N$ by the energy of the state $E_N$ in (52) and (55). Namely, the length of each rod is no longer a fixed number, but is determined by the total deformed energy of the state. This makes the simple linear flow equation (52) into the non-linear inviscid Burgers' equation. We find that similarly for $\lambda > 0$, the deformed energy is always well-defined and approaches to 0 in the $\lambda \to +\infty$ limit. There is a critical value $\lambda_c$ for negative $\lambda$ beyond which the spectrum becomes complex. The break down of the system is tightly related to the shock formation phenomena in Burgers' equation. In fact, the critical value $\lambda_c$ is nothing but wave breaking time, see appendix A for more details.

# 5 Flow equation

In this section, we derive the flow equation for the $T\overline{T}$-deformed finite volume spectrum. We present three derivations for the same flow equation, which shows the consistency of the different approaches and gives us a better understanding of the deformation.

## 5.1   Method 1. Factorization formula

The Lieb-Liniger model can be formulated as a non-relativistic quantum field theory. Similar to the relativistic case, the flow equation can be derived from the factorization formula of the expectation value of the $T\overline{T}$ operator. This has been done in [28], which we recall here. The factorization formula reads[3]

$$\langle n|T\overline{T}|n\rangle = \langle n|T_{00}|n\rangle\langle n|T_{11}|n\rangle - \langle n|T_{01}|n\rangle\langle n|T_{10}|n\rangle. \tag{60}$$

The difference from the relativistic case is that $\langle n|T_{01}|n\rangle \neq \langle n|T_{10}|n\rangle$ since Lorentz invariance is lost. From the definition of the stress energy tensor, we have the following relation

$$\langle n|T_{00}|n\rangle = \frac{E_n(R,\lambda)}{R}, \qquad \langle n|T_{11}|n\rangle = \frac{\partial E_n(R,\lambda)}{\partial R}, \qquad \langle n|T_{10}|n\rangle = \frac{iP_n(R)}{R} \tag{61}$$

The expectation value $\langle n|T_{01}|n\rangle$ in a non-relativistic theory has a more complicated dependence on $P_n$ and $E_n$ and are usually model dependent. We will see that for integrable models which can be solved by Bethe ansatz, it can be expressed in terms of Bethe roots. For the moment, let us denote it by $\langle n|T_{01}|n\rangle = T_n(R,\lambda)/R$. Using the fact that

$$\langle n|T\overline{T}|n\rangle = \frac{1}{R}\partial_\lambda E_n(R,\lambda) \tag{62}$$

We find the following flow equation for the finite volume spectrum

$$\partial_\lambda E_n = E_n\partial_R E_n - \frac{iP_n T_n}{R} \tag{63}$$

## 5.2   Method 2. Bilocal rewriting

The flow equation can also be derived by rewriting the bilinear deformation in terms of the bilocal deformation, as discussed in the previous section. We consider the flow equation for the Hamiltonian (27). It is obvious that the bilocal term does not modify the spectrum because $\langle n|[H, X_{\mathcal{JJ}}]|n\rangle = 0$ for any eigenstate of the Hamiltonian. Using the Feynman-Hellmann theorem, the expectation value obey the following flow equation

$$\partial_\lambda E_n = \langle n|Y_{\mathcal{JJ}}|n\rangle = \langle n|Q_1 j_{Q_2}(0)|n\rangle - \langle n|Q_2 j_{Q_1}(0)|n\rangle \tag{64}$$

Specializing to the $T\overline{T}$ deformation, we take $T_{0a} = \mathcal{J}_1^a$ and $T_{1a} = \mathcal{J}_2^a$, namely $Q_1 = H$, $Q_2 = iP$, $j_{Q_1} = T_{01}$, $j_{Q_2} = T_{11}$. Using the fact that $|n\rangle$ is the eigenstate of $H$ and $P$, we find

$$\partial_\lambda E_n = E_n\langle n|T_{11}|n\rangle - iP_n\langle n|T_{01}|n\rangle = E_n\partial_R E_n - \frac{iP_n T_n}{R} \tag{65}$$

which is the same as (63). Notice that these two derivations are general and does not rely on integrability of the model.

## 5.3   Method 3. Bethe ansatz

Finally we derive the flow equation using Bethe ansatz. This method is quite different from the previous ones and makes use of the integrability machinery. One important benefit is that we can write down an explicit expression for $\langle n|T_{01}|n\rangle$, which cannot be fixed from general considerations.

---

[3]Here we denote the normalized eigenstate by $|n\rangle$ according to the literature. In the Bethe ansatz context, we will denote the eigenstate by $|\mathbf{u}_N\rangle$ to highlight the number of particles and the dependence on rapidities. We use both notations for the normalized eigenstates in this section.

**Bethe ansatz**    The discussion below actually applies to any Bethe ansatz solvable integrable model, so we shall keep the discussions general. The eigenstates of such models can be constructed by Bethe ansatz. Each eigenstate is parameterized by $N$ rapidities $\mathbf{u} = \{u_1, \dots, u_N\}$. We denote the corresponding eigenstate by $|\mathbf{u}_N\rangle$. The energy and momentum of the states are given by

$$H|\mathbf{u}_N\rangle = E_N(\mathbf{u})|\mathbf{u}_N\rangle, \qquad P|\mathbf{u}_N\rangle = P_N(\mathbf{u})|\mathbf{u}_N\rangle \tag{66}$$

where

$$E_N(\mathbf{u}) = \sum_{j=1}^{N} e(u_j), \qquad P_N(\mathbf{u}) = \sum_{j=1}^{N} p(u_j). \tag{67}$$

For non-relativistic continuous quantum mechanical system, the dispersion relation is simply given by

$$e(u) = u^2, \qquad p(u) = u. \tag{68}$$

In the finite volume, the rapidities $\mathbf{u}$ satisfies the Bethe ansatz equations

$$e^{ip(u_j)R} \prod_{k \neq j}^{N} S(u_j, u_k) = 1 \tag{69}$$

where $R$ is the length of the ring and $S(u, v)$ is the factorized two-body S-matrix. In the logarithm form, it is

$$p(u_j)R + \sum_{k \neq j}^{N} \theta(u_j, u_k) = 2\pi I_j, \qquad j = 1, \dots, N \tag{70}$$

where $\theta(u, v) = -i \log S(u, v)$ is the phase shift. Here $I_j$ are momentum mode numbers which can be used to parameterize the Bethe state. The Jacobian matrix between the change from momentum quantum numbers $\{I_j\}_N$ and rapidities $\{u_j\}_N$ is given by the Gaudin matrix whose matrix elements are

$$G_{jk} = 2\pi \frac{\partial I_j}{\partial u_k}, \qquad j, k = 1, \dots, N. \tag{71}$$

Or, more explicitly

$$G_{jk} = \delta_{jk} \left[ p'(u_j)R + \sum_{l=1}^{N} \varphi(u_j, u_l) \right] - \varphi(u_j, u_k) \tag{72}$$

where

$$\varphi(u, v) = -i \frac{\partial}{\partial u} \log S(u, v) \tag{73}$$

is the TBA kernel.

**Deformed Bethe ansatz**   Under the $T\overline{T}$ deformation, the S-matrix is modified according to (46). The deformed BAE in the logarithm form is given by

$$p(u_j)R + \sum_{k \neq j}^{N} \theta_\lambda(u_j, u_k) = 2\pi I_j, \qquad j = 1, \ldots, N \tag{74}$$

where

$$\theta_\lambda(u, v) = \theta(u, v) - \lambda \left[ e(u)p(v) - p(u)e(v) \right]. \tag{75}$$

Equivalently, we can write (74) as

$$p(u_j)[R + \lambda E_N(\mathbf{u})] - \lambda e(u_j)P_N(\mathbf{u}) + \sum_{k \neq j}^{N} \theta(u_j, u_k) = 2\pi I_j \tag{76}$$

Two remarks are in order. First of all, taking the sum of all the above equations, the sum over the $\theta(u_j, u_k)$ terms vanish due to unitarity of the S-matrix. We are left with

$$(R + \lambda E_N(\mathbf{u})) \sum_{j=1}^{N} p(u_j) - \lambda P_N(\mathbf{u}) \sum_{j=1}^{N} e(u_j) = RP_N(\mathbf{u}) = 2\pi \sum_{j=1}^{N} I_j \tag{77}$$

This implies that the total momentum of the system is undeformed by the $T\overline{T}$ deformation, as expected. Secondly, if the sum of the mode numbers is zero, we are in the zero momentum sector. In this sector, the deformed BAE takes the same form as the original one with a deformed length $R_\lambda = R + \lambda E_N(\mathbf{u})$. This is the result that we alluded before when discussing the $O_{0,1}$ deformation.

Therefore, in the deformed theory, to find the spectrum, we need to solve the deformed BAE (76) and then plug in the formula (67). The deformed Bethe roots become $\lambda$ dependent $\{u_j(\lambda)\}$ and this is the only source of the $\lambda$ dependence.

**Mean value of current operators**   To derive the flow equation, we need another important result from integrability, which is the formula for the mean value of current operators. Consider a conserved current $\mathcal{J}^a = (q, j)$ whose *normalized* eigenstate is $|\mathbf{u}_N\rangle$, the eigenvalue of the charge is given by

$$\Lambda(\mathbf{u}) = \sum_{j=1}^{N} h(u_j). \tag{78}$$

It can be proven [29, 30] that the corresponding mean value of the current operator in the same state is given by

$$\langle \mathbf{u}_N | j(x) | \mathbf{u}_N \rangle = \mathbf{e}' \cdot G^{-1} \cdot \mathbf{h}. \tag{79}$$

Here $\mathbf{e}'$ and $\mathbf{h}$ are $N$ dimensional vectors with elements

$$(\mathbf{e}')_j = \frac{\partial e(u_j)}{\partial u_j}, \qquad (\mathbf{h})_j = h(u_j). \tag{80}$$

and $G^{-1}$ is the inverse of the Gaudin matrix (71). In terms of components, we can write

$$\langle \mathbf{u}_N | j(x) | \mathbf{u}_N \rangle = \frac{1}{2\pi} e'(u_j) \frac{\partial u_j}{\partial I_k} h(u_k) \tag{81}$$

In the $T\overline{T}$ deformed theory, we simply replace the Gaudin matrix to the deformed one and take into account the fact that the rapidities that are $\lambda$ dependent. The deformed mean value is given by

$$\langle \mathbf{u}_N | j(x) | \mathbf{u}_N \rangle_\lambda = \mathbf{e}' \cdot G_\lambda^{-1} \cdot \mathbf{h}. \tag{82}$$

**Derivation of the flow equation**  The deformed Gaudin matrix takes the same form as the undeformed one, the only difference is that we replace the TBA kernel to $\varphi_\lambda(u,v)$ where

$$\varphi_\lambda(u,v) = \varphi(u,v) - \lambda[e'(u)p(v) - p'(u)e(v)]. \tag{83}$$

Now let us consider the flow equation for the deformed spectrum. Taking derivative of $E_N(\mathbf{u})$ with respect to $\lambda$.

$$\frac{\partial}{\partial \lambda} E_N(\mathbf{u}) = \frac{\partial}{\partial \lambda} \sum_{j=1}^{N} e(u_j) = \sum_{j=1}^{N} e'(u_j) \frac{\partial u_j}{\partial \lambda} \tag{84}$$

Using the fact that

$$\frac{\partial u_j}{\partial \lambda} = \frac{\partial u_j}{\partial I_k} \frac{\partial I_k}{\partial \lambda} = \frac{1}{2\pi} \frac{\partial u_j}{\partial I_k} \left( p(u_k) E_N(\mathbf{u}) - e(u_k) P_N(\mathbf{u}) \right) \tag{85}$$

we arrive at

$$\frac{\partial}{\partial \lambda} E_N(\mathbf{u}) = E_N(\mathbf{u}) \left( \mathbf{e}' \cdot G^{-1} \cdot \mathbf{p} \right) - P_N(\mathbf{u}) \left( \mathbf{e}' \cdot G^{-1} \cdot \mathbf{e} \right) \tag{86}$$

We see that the quantities in the brackets of the right hand side takes the form of expectation value of current operators (79), which is consistent with the previous two methods. Furthermore, notice that we have

$$\mathbf{e}' \cdot G^{-1} \cdot \mathbf{p} = \partial_R E_N(\mathbf{u}) \tag{87}$$

The flow equation can be written as

$$\frac{\partial}{\partial \lambda} E_N(\mathbf{u}) = E_N(\mathbf{u}) \partial_R E_N(\mathbf{u}) - P_N(\mathbf{u}) \left( \mathbf{e}' \cdot G^{-1} \cdot \mathbf{e} \right) \tag{88}$$

Therefore we find

$$\langle \mathbf{u} | T_{01} | \mathbf{u} \rangle = \mathbf{e}' \cdot G^{-1} \cdot \mathbf{e} \tag{89}$$

In what follows, we will find the deformed Bethe roots in various cases. This gives us the deformed spectrum of all the conserved charges and the expectation values of the corresponding currents.

# 6  Deformed spectrum I. $N$-particle states

In this section, we consider the deformed spectrum for $N$-particle states where $N$ is any finite integer. We will first discuss the zero momentum sector and then move to the generic case. For each case, we first consider the deformed spectrum in the free fermion limit, where analytical results can be found. To have access to finite $c$, we can either perform a $1/c$ expansion at large $c$ or study the spectrum at finite $c$ numerically. The study in free fermion limit is not only useful to learn about the qualitative features of the deformed spectrum, but also provide useful starting points for numerical calculations at finite $c$.

## 6.1 Zero momentum sector

Let us first consider the zero momentum sector. Notice that the ground state belongs to this sector. Taking $P_N = 0$, the flow equation simplifies to the inviscid Burgers' equation

$$\partial_\lambda E_N(R, \lambda) = E_N(R, \lambda)\partial_R E_N(R, \lambda). \tag{90}$$

It has the formal solution

$$E_N(R, \lambda) = E_N(R + \lambda E_N, 0). \tag{91}$$

If we know $E_N(R, 0)$ as a function of $R$ explicitly. The formal solution (91) leads to an algebraic equation, which can be solved to give the deformed spectrum. One well-known example is in 2d CFT where $E_N(R, 0) \sim R^{-1}$ and the deformed spectrum takes a square root form. For generic interacting systems, $E_N(R, 0)$ is a complicated function of $R$, sometimes even impossible to write down explicitly. Therefore in general we need to resort to numerical methods. However, for some special cases we can obtain analytical results, which will be discussed in what follows.

Before we embark on details, let us make an important comment. The inviscid Burgers' equation is well-studied in hydrodynamics. It is known that the solutions of the Burgers' equation tend to develop shocks, see appendix A for more details. Applying this to our current situation, we expect that for $\lambda < 0$ there will be singularities which occur at certain critical value $\lambda_c$ where the deformed spectrum is no longer well-defined. From the experience of relativistic theories, we expect the deformed spectrum becomes complex at this point. This is a general phenomena for the spectrum of $T\overline{T}$ deformed theories. We will confirm this point by explicit calculations. This also matches our intuition learned from the $O_{0,1}$ deformation. The precise value of $\lambda_c$ depends on the coupling.

**The free fermion limit**

We first consider the free fermion limit where $c \to \infty$ where $\theta(u, v) = 0$ and the BAE (76) simplifies to

$$u_j \left(R + \lambda E_N(\mathbf{u})\right) = 2\pi I_j, \qquad j = 1, 2, \ldots \tag{92}$$

The undeformed BAE is almost trivial

$$u_j R = 2\pi I_j, \qquad j = 1, \ldots, N. \tag{93}$$

and the undeformed energy is

$$E_N(R, 0) = \frac{\alpha_N}{R^2}, \qquad \alpha_N = 4\pi^2 \sum_{j=1}^{N} I_j^2. \tag{94}$$

The $N$-particle ground state corresponds to the following choice of $I_j$

$$I_j = -\frac{N+1}{2} + j, \qquad j = 1, \cdots, N. \tag{95}$$

and

$$\alpha_N = \frac{\pi^2}{3}(N^3 - N). \tag{96}$$

We can now use the formal solution (91) to find the deformed spectrum. Denoting $x = E_N(R, \lambda)$, we have the following equation

$$x = \frac{\alpha_N}{(R + \lambda x)^2}. \tag{97}$$

In other words, the deformed energy spectrum is given by the zero of the function

$$f(x) = \lambda^2 x^3 + 2R\lambda x^2 + R^2 x - \alpha_N \tag{98}$$

For $\lambda \neq 0$, this is a cubic polynomial and the equation $f(x) = 0$ has multiple solutions. To determine which solution to take as the deformed spectrum, we need to analyze this function in more detail. There are two extremal points $f'(x) = 0$ located at $x_1 = -R/\lambda$ and $x_2 = -R/(3\lambda)$ and we have

$$f(x_1) = -\alpha_N, \qquad f(x_2) = -\alpha_N - \frac{4R^3}{27\lambda} \tag{99}$$

Notice that $\alpha_N > 0$, so that $f(x_1) < 0$. We have the following two case

- $\lambda > 0$, we have $f(x_2) < f(x_1) < 0$, which implies that $f(x)$ only intercept with the real axis once. So there is only one real root, the other two roots are complex. The plot for a few $f(x)$ is given in the left panel of figure 2.

- $\lambda < 0$, there exhibit a critical value $\lambda_c$ defined by $f(x_2) = 0$. More explicitly,

$$\lambda_c = -\frac{4R^3}{27\alpha_N} \tag{100}$$

For $\lambda < \lambda_c$, we have $f(x_2) > 0$ and there are 3 real solutions, for $\lambda_c < \lambda < 0$ there is one real solution, for $\lambda = \lambda_c$ there are 2 real roots. This is shown in the right panel of figure 2.

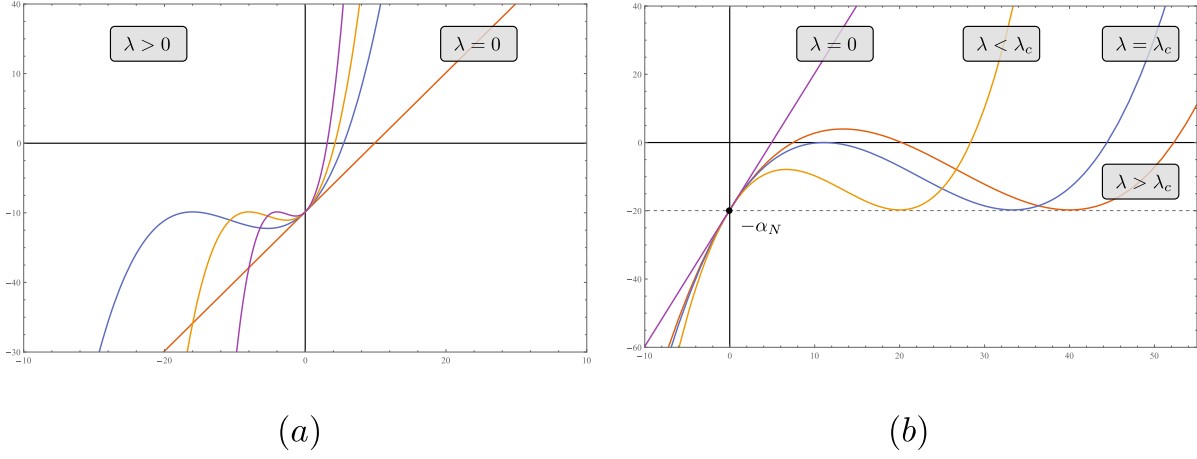

$(a)$ $\qquad\qquad\qquad\qquad\qquad\qquad\qquad$ $(b)$

Figure 2: The plot of $f(x)$ for different values of $\lambda$. We take $R = 2, N = 2$ in the plot. On the left panel, we plot $f(x)$ for different values of $\lambda > 0$. There is only one real root for positive $\lambda$. On the right panel, we consider different values of $\lambda < 0$. There are three cases. For $\lambda_c < \lambda < 0$, there 3 real roots, for $\lambda < \lambda_c$ there is 1 real root, for $\lambda = \lambda_c$ there are 2 real roots.

The cubic equation $f(x) = 0$ can be solved analytically. There are several branches of the solution. To identify a solution as the deformed spectrum, we require that it is regular in the $\lambda \to 0$ limit. In the $\lambda > 0$ regime, there is only one real solution, which turns out to be the branch that is regular in the $\lambda \to 0_+$ limit, and we can naturally identify this root as the deformed energy spectrum. In the regime $\lambda_c < \lambda < 0$, there are three real solutions, but only one of them is regular in the $\lambda \to 0_-$ limit, which we take as the deformed spectrum. For $\lambda < \lambda_c$, the real root corresponds to the branch which diverges at $\lambda = 0$. The rest two roots are complex, therefore the deformed spectrum is no longer well-defined. The qualitative feature of the spectrum is depicted in figure 3. We see that as

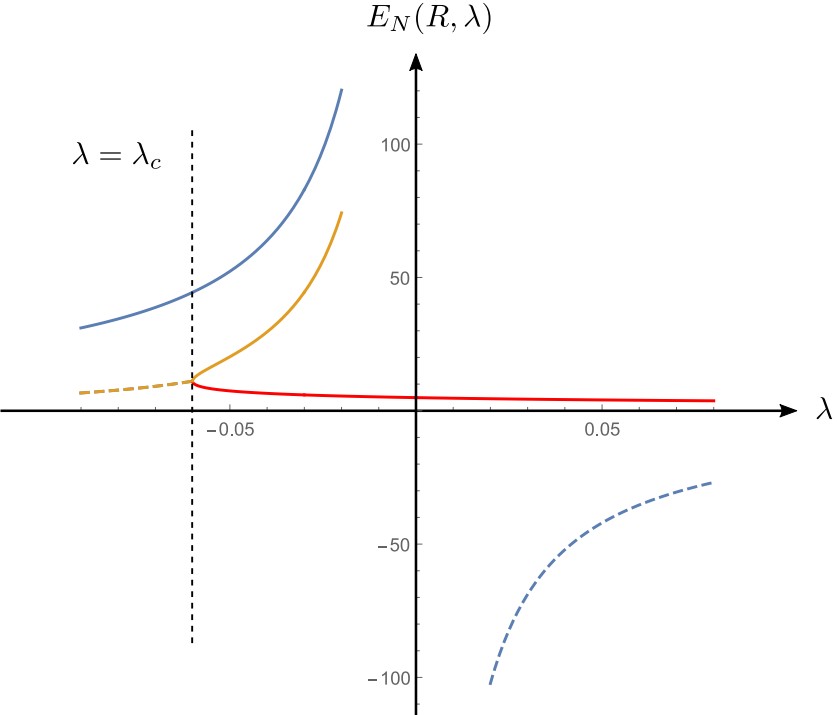

Figure 3: Plot of the zeros of $f(x) = 0$. The solid lines represent the real values, the dashed lines represent the real part of the complex values. We see that in the regime $\lambda \geq \lambda_c$ (the red line) we can identify the deformed spectrum unambiguously.

$\lambda \to \infty$, the energy is decreasing and approach to zero.

In fact, the physical deformed spectrum corresponding to the red line in figure 3 can be written in a compact form as

$$E_N(R, \lambda) = \frac{2R}{3\lambda}\left(\cosh\left[\frac{2}{3}\text{arcsinh}\left(\frac{3\sqrt{3\alpha_N}\sqrt{\lambda}}{2R^{3/2}}\right)\right] - 1\right) \tag{101}$$

$$= \frac{2R}{3\lambda}\left(\cosh\left[\frac{2}{3}\text{arcsinh}\left(\frac{3}{2}\sqrt{\frac{3\lambda E_N(R,0)}{R}}\right)\right] - 1\right)$$

The deformed energy $E_N(R, \lambda)$ in (101) is regular at $\lambda = 0$ and allows a well-defined perturbative expansion

$$E_N(R, \lambda) = \frac{\alpha_N}{R^2} - \frac{2\alpha_N^2}{R^5}\lambda + \frac{7\alpha_N^3}{R^8}\lambda^2 - \frac{30\alpha_N^4}{R^{11}}\lambda^3 + \cdots \tag{102}$$

The function $E_N(R, \lambda)$ is monotonically decreasing and approaches to zero as $\lambda \to \infty$. For $\lambda < 0$, $E_N(R, \lambda)$ is real when

$$\sqrt{|\lambda|} \leq \frac{2R^{3/2}}{3\sqrt{3}\sqrt{\alpha_N}} \quad \Leftrightarrow \quad -\frac{4R^3}{27\alpha_N} = \lambda_c \leq \lambda < 0 \tag{103}$$

which is the same critical value as we obtained before. This behavior is what we have expected physically. For $\lambda > 0$, the space between the particles is increased and in the limit $\lambda \to +\infty$ they are so widely separated and almost do not interact, wich trivializes the model in this limit. On the other hand, for $\lambda < 0$, we have the hard rod picture. Now the size of the rod is determined by the deformed energy of the state, which can be found by solving the Burgers' equation.

To sum up, the deformed energy is well-defined and monotonically decreasing to zero in the regime $\lambda \geq \lambda_c$. The largest deformed energy is achieved at $\lambda = \lambda_c$ and is given by

$$E_N(R, \lambda_c) = \frac{9\alpha_N}{4R^2} = \frac{9}{4}E_N(R, 0). \tag{104}$$

Let us compare this spectrum with the relativistic case. In the zero momentum sector of deformed CFTs, we need to solve the algebraic equation of the form $E_n = \beta_n/(R + \lambda E_n)$. This is easily solved and the result is given by

$$E_n(R, \lambda) = \frac{1}{2\lambda}\left(\sqrt{R^2 + 4\lambda\beta_n} - R\right) \tag{105}$$

For states with $\beta_n > 0$, we have the same qualitative feature. For $\lambda > 0$, the function is real and monotonically decreasing; for $\lambda < 0$ we have a critical value at $\lambda_c = -R^2/(4\beta_n)$. In the regime $\lambda \geq \lambda_c$ the deformed energy is real and well-defined. For $\lambda < \lambda_c$ the deformed energy becomes complex.

**The deformed Bethe roots** After finding the deformed spectrum, we can find the deformed Bethe roots (106)

$$u_j(\lambda) = \frac{2\pi I_j}{R + \lambda E_N(R, \lambda)}. \tag{106}$$

We see that the only difference is that now the radius for the quantization condition is given by $R_\lambda = R + \lambda E_N(R, \lambda)$. We focus on the regime $\lambda \geq \lambda_c$ where the deformed spectrum is well define. In this regime, the radius $R_\lambda$ is increasing monotonically. The smallest radius is reached at $\lambda = \lambda_c$

$$R_c = R + \lambda_c E_N(R, \lambda_c) = \frac{2}{3}R. \tag{107}$$

The Bethe roots contain all the information of the state. For example, we can compute the deformed conserved charges using the Bethe roots.

Before ending this subsection, let us make the following comment. We see that in the $T\overline{T}$ deformed case, the critical value of $\lambda_c$ does not occur at $R_\lambda = 0$. This is due to the non-linearity of the flow equation in this case. Although the hard rod intuition is still correct, but now the size of the rod is no longer a simple fixed value, but need to be found by solving the flow equation.

**Perturbation theory**

We now move beyond the free fermion point. We consider the $1/c$ expansion in the large $c$ limit in this subsection. We have

$$\theta(u,v) = \frac{2(u-v)}{c} - \frac{2}{3}\frac{(u-v)^3}{c^3} + \mathcal{O}(c^{-5}). \tag{108}$$

Let us denote the deformed energy which depend on $c$ as $\mathcal{E}_N(R,\lambda;c)$ and expand it in terms of $1/c$

$$\mathcal{E}_N(R,\lambda;c) = \sum_{k=0}^{\infty} \frac{E_N^{(k)}(R,\lambda)}{c^k} \tag{109}$$

where $E_N^{(0)}(R,\lambda) = E_N(R,\lambda)$ is the deformed energy in the free fermion limit. Each term in the expansion is non-perturbative in $\lambda$. In the zero momentum sector, the deformed spectrum satisfies the Burgers' equation and we have

$$\mathcal{E}_N(R,\lambda,;c) = \mathcal{E}_N(R+\lambda\mathcal{E}_N,0;c) \tag{110}$$

Therefore we first determine the undeformed energy $\mathcal{E}_N(R,0;c)$ and find out its dependence on $R$. Then we solve the algebraic equation (110) to find the deformed energy. The result for the first two corrections are given by

$$E_N^{(1)} = -\frac{4N\alpha_N}{R_\lambda^3 + 2\lambda\alpha_N}, \tag{111}$$

$$E_N^{(2)} = \frac{4N^2\alpha_N(R_\lambda^6 - 8\lambda\alpha_N R_\lambda^3 - 8\lambda^2\alpha_N^2)}{R_\lambda(R_\lambda^3 + 2\lambda\alpha_N)^3}.$$

where

$$R_\lambda = R + \lambda E_N(R,\lambda). \tag{112}$$

with $\alpha_N$ given in (94) and $E_N(R,\lambda)$ given in (101).

**Numerical results**

Finally we consider the deformed spectrum at finite $c$. The BAE to solve is (76)

$$p(u_j)[R + \lambda E_N(\mathbf{u})] + \sum_{k\neq j}^{N} \theta(u_j, u_k) = 2\pi I_j, \qquad j = 1,\ldots,N \tag{113}$$

where at finite $c$ we have

$$\theta(u,v) = 2\arctan\left(\frac{u-v}{c}\right). \tag{114}$$

The equation (113) can only be solved numerically. Our numerical strategy is as follows. We first solve the equation at the free fermion limit $c \to \infty$ where analytical results are known. This provides a 'seed' solution for the BAE. Then we find solutions for finite $c$ by iterations. Two comments are in order. Firstly, one might try to first find the solution at finite $c$ and $\lambda = 0$, and then by varying $\lambda$ to find the deformed Bethe roots. This approach is less stable numerically, due to the fact that there are multiple solutions to the deformed BAE. Secondly, when trying to find the deformed Bethe roots, we should work in the regime $\lambda > \lambda_c$, otherwise the iteration procedure becomes numerically unstable. The critical value $\lambda_c$ at finite $c$ can be determined numerically. In most cases, it is sufficient to take the critical value at the free fermion limit. We present some numerical results below.

**Deformed spectrum**   As an example, we consider the deformed spectrum for the ground state for $N = 10$, $R = 30$. The critical value of $\lambda$ at the free fermion limit is $\lambda_c \approx -1.22814$. The deformed energy for different value of $c$ is presented in figure 4. To see the dependence of $\mathcal{E}_N(R, \lambda, c)$ on $c$

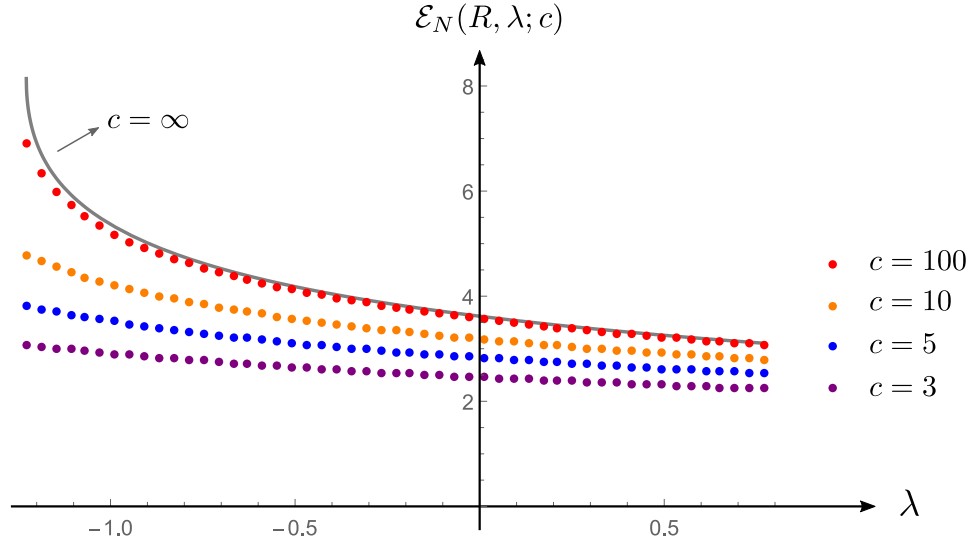

Figure 4: Deformed energy $\mathcal{E}_N(R, \lambda, c)$ for finite $c$. We take $N = 10$, $R = 30$ and plot the energy from $\lambda_c = -1.22814$ to $\lambda_c + 2$ for $c = 3, 5, 10, 100$. The gray continuous line is the deformed energy at $c = \infty$, which is given by the analytical result. We see the deformed energy decreases while decreasing the value of $c$.

and $\lambda$ more explicitly, we present the plot of the deformed spectrum for different values of $c$ and $\lambda$ in figure 5. We see clearly that the deformed energy increases (decreases) while increasing $c$ ($\lambda$).

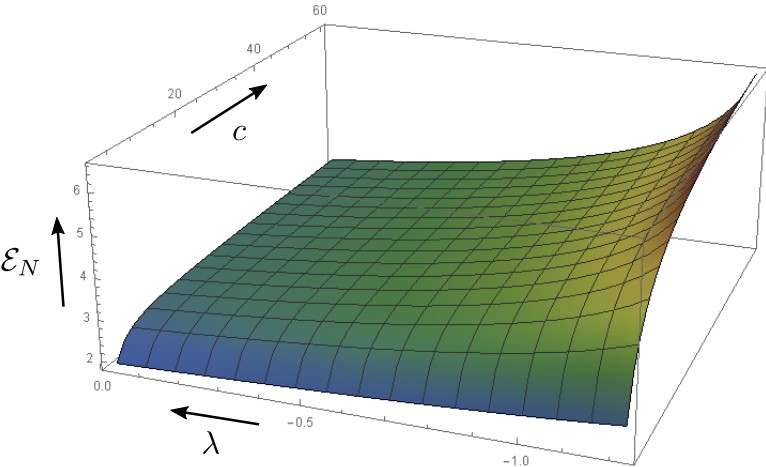

Figure 5: Deformed spectrum for different values of $c$ and $\lambda$. We see that the spectrum increases as we decrease $\lambda$ and increase $c$.

**Deformed Bethe roots**   Let us now discuss how does the deformation affects the distribution of Bethe roots. At the free fermion limit, we simply replace the length by $R \to R_\lambda = R + \lambda E_N$. For

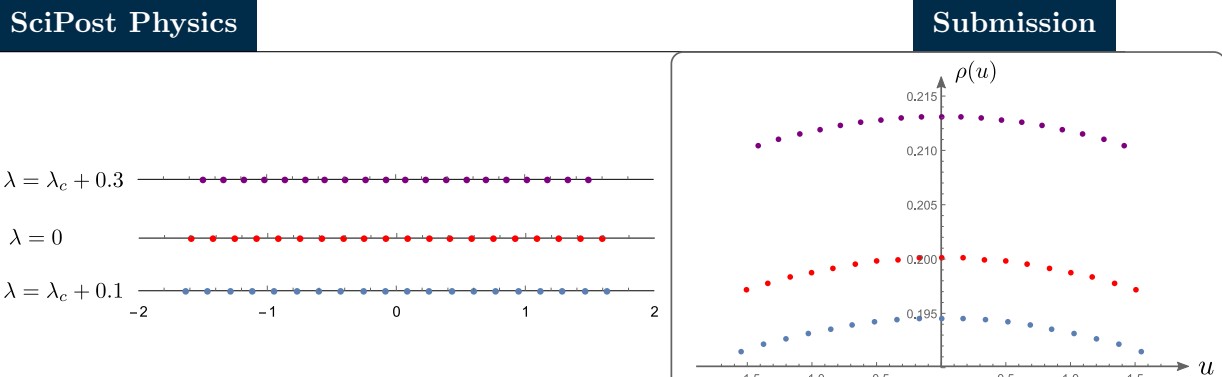

Figure 6: Distribution of Bethe roots for $N = 20$, $R = 30$ and $c = 5$. We plot the root distribution for three different $\lambda$, where we take $\lambda_c$ to be the critical point of the free fermion point. The left panel gives the position of the roots on the real axis. The right panel shows the corresponding densities of the rapidities. We see clearly the Bethe roots are more and more condensed while increasing $\lambda$.

positive $\lambda$, $R_\lambda > R$ and the Bethe roots tend to be more densely distributed on the real axis. For $\lambda_c \leq \lambda < 0$, the situation is the opposite. Going away from the free fermion point, the qualitative feature is the same. We present the distribution of the Bethe roots for $N = 10$, $R = 30$, $c = 5$ at different values of $\lambda$ in figure 6.

## 6.2 Generic states

In this subsection, we consider more general states where the momentum is non-zero. The Bethe equation reads

$$u_j \left[ R + \lambda E_N(\lambda) \right] - \lambda u_j^2 P_N + \sum_{k \neq j}^{N} \theta(u_j, u_k) = 2\pi I_j, \qquad j = 1, \ldots, N. \tag{115}$$

As before, we first consider the Girardeau-Tonks limit where $\theta(u, v) \to 0$. Then each equation becomes quadratic in $u_j$

$$u_j \left[ R + \lambda E_N(\lambda) \right] - \lambda u_j^2 P_N = 2\pi I_j, \qquad j = 1, \ldots, N. \tag{116}$$

Solving these equations, we find

$$u_j = \frac{R + \lambda E_N(\lambda) - \sqrt{[R + \lambda E_N(\lambda)]^2 - 8\pi I_j \lambda P_N}}{2\lambda P_N} \tag{117}$$

To find the deformed energy, we can solve the following algebraic equation

$$P_N = \sum_{j=1}^{N} u_j = \sum_{j=1}^{N} \frac{R + \lambda E_N(\lambda) - \sqrt{[R + \lambda E_N(\lambda)]^2 - 8\pi I_j \lambda P_N}}{2\lambda P_N} \tag{118}$$

This is much harder to solve analytically, it is not clear whether closed form expression exists for any interger $N$. We can find the analytical result by a formal series expansion in $\lambda$ or solve the

system numerically. To this end, let us expand $E_N(\lambda)$ as

$$E_N(\lambda) = \sum_{k=0}^{\infty} \mathrm{E}_N^{(k)} \lambda^k \tag{119}$$

Notice that $P_N$ is not deformed and does not depend on $\lambda$. Plug (119) in (118) we can find $\mathrm{E}_N^{(k)}$ order by order. To write the result in a more compact way, we introduce the following notation

$$M_s = \sum_{j=1}^{N} \left(\frac{2\pi I_j}{R}\right)^s \tag{120}$$

It is clear that

$$P_N = M_1, \qquad \mathrm{E}_N^{(1)} = M_2. \tag{121}$$

The first few $\mathrm{E}_N^{(k)}$ are given by

$$\mathrm{E}_N^{(1)} = \frac{1}{R}(-M_2^2 + 2M_1 M_3), \tag{122}$$

$$\mathrm{E}_N^{(2)} = \frac{1}{R^2}(7M_2^3 - 12M_1 M_2 M_3 + 5M_1^2 M_4),$$

$$\mathrm{E}_N^{(3)} = \frac{1}{R^3}(-30M_2^4 + 72M_1 M_2^2 M_3 - 16M_1^2 M_3^2 - 40M_1^2 M_2 M_4 + 14M_1^3 M_5).$$

Taking $M_1 = 0$, we indeed recover the zero momentum result. The perturbative expansion result is also useful for finding numerical solutions of (118) as it provides seed solutions for the deformed BAE. We can follow a similar strategy to go beyond the free fermion limit. Here we have to do most calculations perturbatively and numerically. The qualitative features are similar and we will not repeat the analysis here.

## 6.3 Other deformed quantities

In this subsection, we consider the deformation of other quantities under the $T\overline{T}$ deformation. In particular, we are interested in the deformed conserved charges and the average values of the current operators and the effective velocity.

**Deformed conserved charges** The eigenvalue of the deformed conserved charge $Q_a$ is given by

$$\Lambda_a(\mathbf{u}) = \sum_{k=1}^{N} h_a(u_k) \tag{123}$$

We can write down the flow equation for the charge

$$\partial_\lambda \Lambda_a(\mathbf{u}) = E_N(\mathbf{u}) \left(\mathbf{h}'_a \cdot G^{-1} \cdot \mathbf{p}\right) - P_N \left(\mathbf{h}'_a \cdot G^{-1} \cdot \mathbf{e}\right) \tag{124}$$

The quantities in the bracket is the expectation value of the *generalized current operator*. Consider two conserved charges $Q_a$, $Q_b$ whose corresponding charge and current densities are $(q_a(x), j_a(x))$ and $(q_b(x), j_b(x))$ respectively. The generalized current $J_a^b(x)$ is defined by

$$i[Q_b, q_a(x)] = \partial_x J_a^b(x). \tag{125}$$

It is proven in [29, 30] that the expectation value of generalized current density is

$$\langle \mathbf{u}_N | J_a^b(x) | \mathbf{u}_N \rangle = \mathbf{h}_b' \cdot G^{-1} \cdot \mathbf{h}_a. \tag{126}$$

Therefore the flow equation (124) implies the following deformation for the conserved charges

$$\frac{\mathrm{d}}{\mathrm{d}\lambda} Q_a = \int \left( h(x) j_a^P(x) - p(x) j_a^H(x) \right) \mathrm{d}x. \tag{127}$$

under $T\overline{T}$ deformation.

**Expectation value of current operator**    Another interesting quantity is the deformation of the mean value of the current density operator. This is related to another important quantity called the *effective velocity*. Recall the formula for the expectation value of the current (81)

$$\langle \mathbf{u}_N | j_a(x) | \mathbf{u}_N \rangle = \frac{1}{2\pi} e'(u_j) \frac{\partial u_j}{\partial I_k} h(u_k) = \frac{1}{R} \sum_{k=1}^{N} v_{\mathrm{eff}}(u_k) h_a(u_k). \tag{128}$$

where we have defined the effective velocity $v_{\mathrm{eff}}$ of the particle with $u_k$ as

$$v_{\mathrm{eff}}(u_k) = \frac{R}{2\pi} \frac{\partial E}{\partial I_k} = \frac{R}{2\pi} \sum_{j=1}^{N} 2u_j \frac{\partial u_j}{\partial I_k}. \tag{129}$$

Notice that $\partial(2\pi I_j)/\partial u_k$ is the matrix elements of Gaudin matrix, we can write $v_{\mathrm{eff}}(u_k)$ as

$$v_{\mathrm{eff}}(u_k) = 2R \sum_{j=1}^{N} u_j (G^{-1})_{jk} \tag{130}$$

The effective velocity describes how fast the particle moves in the presence of other particles. To gain more intuitions about it, let us consider the effective velocities of one- and two-particle states. For one-particle states, the Gaudin matrix is trivial, and we have $(G^{-1}) = 1/R$. Therefore

$$v_{\mathrm{eff}}(u_1) = 2u_1 = \frac{e'(u_1)}{p'(u_1)} = \frac{\partial e}{\partial p} \tag{131}$$

which means the one-particle effective velocity is nothing but the usual group velocity of the particle. The bilinear deformation does not deform the one-particle state, therefore the one-particle effective velocity is not modified.

For the two-particle state, the Gaudin matrix reads

$$G = \begin{pmatrix} R + \varphi_{12} & -\varphi_{12} \\ -\varphi_{21} & R + \varphi_{21} \end{pmatrix} \tag{132}$$

We have

$$G^{-1} = \frac{1}{\det G} \begin{pmatrix} R + \varphi_{21} & \varphi_{12} \\ \varphi_{21} & R + \varphi_{12} \end{pmatrix} \tag{133}$$

The effective velocities are

$$v_{\mathrm{eff}}(u_1) = \frac{2Ru_1 + 2(u_1 + u_2)\varphi_{21}}{R + \varphi_{12} + \varphi_{21}}, \tag{134}$$

$$v_{\mathrm{eff}}(u_2) = \frac{2Ru_2 + 2(u_1 + u_2)\varphi_{12}}{R + \varphi_{12} + \varphi_{21}}.$$

For simplicity, we consider the ground state where $u_1 + u_2 = 0$, $\varphi_{ij} = \varphi_{ji}$. In this case,

$$v_{\text{eff}}(u_1) = \frac{2u_1}{1 + 2\varphi_{12}/R}, \qquad v_{\text{eff}}(u_2) = \frac{2u_2}{1 + 2\varphi_{12}/R} \tag{135}$$

We see that in the free fermion limit the effective velocity is the same as the group velocity. Now we consider the $T\overline{T}$ deformation. We replace $u_j \to u_j(\lambda)$ and $\varphi_{ij} \to \varphi_{ij}^\lambda$. Notice that after the deformation, the TBA kernel is no longer symmetric, namely $\varphi_{ij}^\lambda \neq \varphi_{ji}^\lambda$. We have

$$v_{\text{eff}}(u_1) = \frac{2u_1(\lambda)}{1 + (\varphi_{12}^\lambda + \varphi_{21}^\lambda)/R}, \qquad v_{\text{eff}}(u_2) = \frac{2u_2(\lambda)}{1 + (\varphi_{12}^\lambda + \varphi_{21}^\lambda)/R} \tag{136}$$

For the ground state in the free fermion limit, we have $v_{\text{eff}}(u_2) = -v_{\text{eff}}(u_1)$ and

$$v_{\text{eff}}(u_1) = \frac{2Ru_1(\lambda)}{R + 6\lambda u_1(\lambda)^2} \tag{137}$$

Recall from (106), we have

$$u_1(\lambda) = \frac{\pi}{R + \lambda E_2(R, \lambda)}. \tag{138}$$

The critical value for $\lambda$ in this case is $\lambda_c = -2R^3/(27\pi^2)$. We can check explicitly that

$$\lim_{\lambda \to \lambda_c} v_{\text{eff}}(u_i) = \infty. \tag{139}$$

Namely, at the critical value, the effective velocity diverges. On the other hand, in the limit $\lambda \to +\infty$, the effective velocity tends to zero.

The divergence of the effective velocity at the critical value is general. This can be proved by considering $\partial_R E_N(R, \lambda)$. Viewing $E_N(R, \lambda)$ as the solution of the inviscid Burgers' equation. The shock wave formation is characterized by the divergence of $\partial_R E_N(R, \lambda)$. Namely, at the critical value, $\partial_R E(R, \lambda)$ diverges. On the other hand, we can write

$$\partial_R E_N(R, \lambda) = \sum_{k=1}^N v_{\text{eff}}(u_k) p(u_k). \tag{140}$$

At critical value, $p(u_k)$ is finite. Therefore $v_{\text{eff}}(u_k)$ must diverge. This divergence is again consistent with the hard rod picture since the hard rods are so close to each other, the effects from interactions on the velocity becomes extremely strong. It then follows that all the deformed mean value of current densities diverges at the critical value.

# 7 Deformed spectrum II. Thermodynamic limit

In this section, we consider the deformed spectrum in the thermodynamic limit where $R \to \infty$ at zero temperature. The finite temperature case will be studied in the next section. In the thermodynamic limit, it is convenient to introduce the density of Bethe roots $\rho(u)$, defined by

$$\rho(u_j) = \lim_{R, N \to \infty} \frac{1}{R(u_{j+1} - u_j)} \tag{141}$$

Following the standard procedure, the Bethe equation[4] can be as a linear integral equation of the density $\rho(u)$

$$2\pi\rho(u) = 1 + \int_a^b \varphi(u,v)\rho(v)\mathrm{d}v \tag{142}$$

The integration range in (142) is determined by the normalization

$$\int_a^b \rho(u)\mathrm{d}u = \frac{N}{R} \equiv n_0. \tag{143}$$

Using the density of roots, we can rewrite the following type of summation in terms of an integral

$$\sum_{k=1}^N f(u_k) = R\int_a^b f(u)\rho(u)\mathrm{d}u \tag{144}$$

The energy and momentum in the thermodynamic limit are given by

$$\frac{E_N(\mathbf{u})}{R} = \mathbb{E} \equiv \int_a^b u^2\rho(u)\mathrm{d}u, \qquad \frac{P_N}{R} = \mathbb{P} \equiv \int_a^b u\rho(u)\mathrm{d}u. \tag{145}$$

Now we consider the effect of $T\overline{T}$ deformation. We simply deform the TBA kernel in (142) by $\varphi(u,v) \mapsto \varphi_\lambda(u,v)$. The deformed equation can be written as

$$2\pi\rho_\lambda(u) = 1 + \int_{a_\lambda}^{b_\lambda} \varphi_\lambda(u,v)\rho_\lambda(v)\mathrm{d}v \tag{146}$$

$$= 1 - \lambda(2u\mathbb{P} - \mathbb{E}(\lambda)) + \int_{a_\lambda}^{b_\lambda} \varphi(u,v)\rho_\lambda(v)\mathrm{d}v.$$

Notice that the integral range is deformed because $\rho_\lambda(u)$ is in general different from $\rho(u)$ while we still impose the normalization condition

$$\int_{a_\lambda}^{b_\lambda} \rho_\lambda(u)\mathrm{d}u = \frac{N}{R}. \tag{147}$$

## 7.1 The free fermion limit

To gain some intuition about the deformed spectrum, we consider the free fermion limit. In this limit, the TBA kernel vanishes and we have the simple equation

$$2\pi\rho_\lambda(u) = 1 - \lambda(2u\mathbb{P} - \mathbb{E}(\lambda)). \tag{148}$$

This equation can be brought to the form of Fredholm equation of the second kind with *degenerate kernel* and can be solved analytically. Using the method in appendix C, we find the deformed energy and momentum are

$$\mathbb{P} = \frac{\pi M_1}{2\pi^2 + \pi\lambda M_2 + (M_1 M_3 - M_2^2)\lambda^2}, \tag{149}$$

$$\mathbb{E} = \frac{\pi M_2 + \lambda(M_2^2 - M_1 M_3)}{2\pi^2 + \pi\lambda M_2 + (M_1 M_3 - M_2^2)\lambda^2}$$

---

[4]More precisely, the derivative of the Bethe equation.

where the quantities are defined

$$M_k \equiv \int_{a_\lambda}^{b_\lambda} u^k \, \mathrm{d}u. \tag{150}$$

In the zero momentum sector, the integration range is symmetric with respect to the imaginary axis and we have

$$M_k = \int_{-B_\lambda}^{B_\lambda} u^k \mathrm{d}u. \tag{151}$$

In this case, $M_{2k+1} = 0$ and the results simplify to

$$\mathbb{P} = 0, \qquad \mathbb{E}(\lambda) = \frac{M_2}{2\pi - \lambda M_2}. \tag{152}$$

The density of Bethe roots is

$$\rho_\lambda(u) = \frac{1}{2\pi} \left( 1 + \frac{\lambda M_2}{2\pi - \lambda M_2} \right) = \frac{3}{6\pi - 2\lambda B_\lambda^3} \tag{153}$$

We see that the deformed density in the free fermion limit is still uniform, but the range of the integral is changed. To fix $B_\lambda$, we impose the normalization

$$\int_{-B_\lambda}^{B_\lambda} \rho_\lambda(u) \mathrm{d}u = \frac{3B_\lambda}{3\pi - \lambda B_\lambda^3} = n_0. \tag{154}$$

A closed form solution with regular $\lambda \to 0$ limit can be found

$$B_\lambda = \frac{2}{\sqrt{n_0 \lambda}} \sinh \left[ \frac{1}{3} \mathrm{arcsinh} \left( \frac{3\pi}{2} n_0 \sqrt{n_0 \lambda} \right) \right] \tag{155}$$

For $\lambda > 0$, $B_\lambda$ is monotonically decreasing function of $\lambda$ and approaches to 0 as $\lambda \to +\infty$. For $\lambda < 0$, there is a critical value $\lambda_c$ given by

$$\lambda_c = -\frac{9}{4} n_0^3 \pi^2 \tag{156}$$

Within the region $\lambda_c < \lambda < 0$ the function $B_\lambda$ takes real values and are monotonically decreasing. In the region $\lambda < \lambda_c$, $B_\lambda$ becomes complex. Therefore we find that in the region $\lambda > \lambda_c$, $B_\lambda$ is well-defined and real. Using this, we can obtain the explicit expression for all the conserved charges in this region. For example, the energy is given by

$$\mathbb{E}(\lambda) = \frac{2}{\lambda} \left( \cosh \left[ \frac{2}{3} \mathrm{arcsinh} \left( \frac{3\pi}{2} n_0 \sqrt{n_0 \lambda} \right) \right] - 1 \right) \tag{157}$$

This is consistent with what we have found in the $N$-particle state. The density of roots in this case is given by

$$\rho_\lambda(u) = \frac{1}{2\pi} (1 + \lambda \mathbb{E}(\lambda)) = \frac{1}{\pi} \cosh \left[ \frac{2}{3} \mathrm{arcsinh} \left( \frac{3\pi}{2} n_0 \sqrt{n_0 \lambda} \right) \right] - \frac{1}{2\pi} \tag{158}$$

which is a uniform distribution that depends on the parameter $\lambda$. To find the result for finite $c$, we can perform perturbative analysis in $1/c$ or numerical approaches, which parallel what we have done in the $N$-particle state cases.

## 7.2 Zero temperature thermodynamics

In this subsection, we consider the thermodynamics at zero temperature. To this end, it is useful to define the pseudo-energy $\varepsilon(u)$ as follows

$$\varepsilon(u) - \frac{1}{2\pi} \int_{-B}^{B} \varphi(u,v)\varepsilon(v)\mathrm{d}v = u^2 - \mu. \tag{159}$$

The pseudo-energy is also an important quantity for the study of thermodynamics at finite temperature and our definition (159) arises naturally in the zero temperature limit of the TBA equation. The function $\varepsilon(u)$ is an even function that is defined in $|u| \leq B$, the quantity $\mu$ is the chemical potential, which is chosen in such a way that $\varepsilon(u)$ vanishes at the end points of the integration range, namely $\varepsilon(\pm B) = 0$. Using the pseudo-energy and the thermodynamics relation

$$E_0 = -\mathrm{P}R + \mu N \tag{160}$$

we find the pressure of the system is given by

$$\mathrm{P} = -\frac{1}{2\pi} \int_{-B}^{B} \mathrm{d}u\, \varepsilon(u). \tag{161}$$

Now we consider the $\mathrm{T\overline{T}}$ deformation. For simplicity, we consider the free fermion limit

$$\varepsilon_\lambda(u) + \frac{\lambda}{2\pi} \int_{-B_\lambda}^{B_\lambda} (2uv - v^2)\varepsilon_\lambda(v)\mathrm{d}v = u^2 - \mu. \tag{162}$$

This equation can be solved by the method in appendix C. Physically we expect the quantity $\varepsilon_\lambda(u)$ is an even function of $u$. Therefore the equation simplifies to

$$\varepsilon_\lambda(u) - \frac{\lambda}{2\pi} \int_{-B_\lambda}^{B_\lambda} v^2 \varepsilon_\lambda(v)\mathrm{d}v = u^2 - \mu. \tag{163}$$

We search for a solution of the form

$$\varepsilon_\lambda(u) = u^2 - \left(\mu - \lambda A\right) \tag{164}$$

where $A$ is some constant which depends on $\lambda$. The self-consistency relation for $A$ is given by

$$A = \frac{1}{2\pi} \int_{-B_\lambda}^{B_\lambda} u^2(u^2 - \mu - \lambda A)\mathrm{d}u. \tag{165}$$

which can be solved easily and gives

$$A = \frac{3B_\lambda^5 - 5\lambda\mu B_\lambda^3}{15\pi + 5\lambda B_\lambda^3}. \tag{166}$$

If we fix the density $n_0$ and $\lambda$. The integration range has been determined in (155) in terms of $n_0$ and $\lambda$. Plugging in (166) and (164), we find the deformed pseudo-energy. By requiring $\varepsilon_\lambda(\pm B_\lambda)$, we obtain $\mu$ in terms of $B_\lambda$

$$\mu = \frac{1}{5}B_\lambda^2 \left(4 + \frac{3\pi}{3\pi + 2\lambda B_\lambda^3}\right) \tag{167}$$

Plugging into the pseudo-energy, we find

$$\varepsilon_\lambda(u) = u^2 - B_\lambda^2. \tag{168}$$

The pressure of the deformed system is given by

$$\mathrm{P}_\lambda = -\frac{1}{2\pi} \int_{-B_\lambda}^{B_\lambda} (u^2 - B_\lambda^2)\mathrm{d}u = \frac{2}{3\pi} B_\lambda^3. \tag{169}$$

We see that the deformed pressure has a similar behavior as function $B_\lambda^3$ as described in the previous section.

## 8  Finite temperature thermodynamics

In this section, we consider the deformed model at finite temperature and study thermodynamics by the method of thermodynamic Bethe ansatz (TBA) [14]. For an introduction of the TBA approach, we also refer to the books [23, 24]. Here we only write down the key formulas from this approach. The central equation is the TBA equation which can be derived from Bethe equations together with thermodynamics. It is a non-linear integral equation of the pseudo-energy $\varepsilon(u)$

$$\varepsilon(u) = u^2 - \mu - \frac{T}{2\pi} \int_{-\infty}^{\infty} \varphi(u, v) \ln\left(1 + e^{-\varepsilon(v)/T}\right) \mathrm{d}v \tag{170}$$

Here $T$ and $\mu$ are the temperature and chemical potential, respectively. $\varphi(u, v)$ is the TBA kernel of the model. The pseudo-energy is defined by

$$\frac{\rho_h(u)}{\rho(u)} = e^{\varepsilon/T} \tag{171}$$

where $\rho_h$ and $\rho$ are the densities of holes and particles. Combined with the equation

$$2\pi\rho(u)(1 + e^{\beta\varepsilon(u)}) = 1 + \int_{-\infty}^{\infty} \varphi(u, v)\rho(v)\mathrm{d}v \tag{172}$$

we can determine the densities $\rho(u)$ and $\rho_h(u)$. In practice, we need to solve the TBA equation either analytically or numerically to find $\varepsilon(u)$. The free energy of the system is given in terms of the pseudo-energy as

$$F = -T \ln Z = N\mu - \frac{TR}{2\pi} \int_{-\infty}^{\infty} \ln\left(1 + e^{-\varepsilon(u)/T}\right) \mathrm{d}u. \tag{173}$$

Thermodynamic quantities can be obtained from the free energy. For example, the pressure of the system is given by

$$\mathrm{P} = -\frac{\partial F}{\partial R} = \frac{T}{2\pi} \int_{-\infty}^{\infty} \ln\left(1 + e^{-\varepsilon(u)/T}\right) \mathrm{d}u. \tag{174}$$

In what follows, we will always consider the pressure P as the main quantity to study. Other quantities can be studied in a similar way. The TBA equation (170) cannot be solved analytically in general. In order to gain some intuitions, we will first consider the $O_{0,1}$ deformation in the free fermion limit as a warm-up. As in the spectral problem, this simpler case already captures some salient features of the $T\overline{T}$ deformation. Then we consider the free boson and free fermion limits of the deformed Lieb-Liniger model. From the solution of these two limiting cases, it is straightforward to generalize to the generic $c$ case.

## 8.1 The hard rod gas

We first consider the $O_{0,1}$ deformed Lieb-Liniger model in the free fermion limit, or equivalently the hard rod model[5]. The TBA kernel is simply a constant $\varphi(u,v) = -a$ where $a > 0$ is the length of the hard rod. The main equations (170) and (174) become

$$\varepsilon(u) = u^2 - \mu + \frac{a}{2\pi\beta} \int_{-\infty}^{\infty} \ln\left(1 + e^{-\varepsilon(v)/T}\right) dv \tag{175}$$

and the pressure P is given by

$$\mathrm{P} = \frac{1}{2\pi\beta} \int_{-\infty}^{\infty} \ln\left(1 + e^{-\beta\varepsilon(u)}\right) du. \tag{176}$$

From these equations, we see that the quasi-energy $\varepsilon(u)$ takes the form

$$\varepsilon(u) = u^2 - \mu + a\mathrm{P}. \tag{177}$$

Comparing to the free fermion case, we see that the effect of the finite size is shifting the chemical potential by $-a\mathrm{P}$. The value of this shift can be determined by the self-consistency relation by plugging (177) into (176)

$$\mathrm{P} = \frac{1}{2\pi\beta} \int_{-\infty}^{\infty} \ln\left(1 + e^{-\beta(u^2 - \mu + a\mathrm{P})}\right) du \tag{178}$$

Using the formula in appendix D, we have

$$\mathrm{P} = -\frac{1}{2\sqrt{\pi}\beta^{3/2}} \mathrm{Li}_{\frac{3}{2}}\left(-e^{\eta}\right) = \frac{1}{2\sqrt{\pi}\beta^{3/2}} \mathcal{F}_{\frac{1}{2}}(\eta), \qquad \eta = \beta(\mu - a\mathrm{P}) \tag{179}$$

where $\mathcal{F}_s(\eta)$ is the Fermi-Dirac integral. Let us define a function

$$g_0(x) = \frac{1}{2\sqrt{\pi}\beta^{3/2}} \mathcal{F}_{\frac{1}{2}}(\beta\mu - a\beta x) \tag{180}$$

The value is P is determined at $x = g_0(x)$. A plot for $g_0(x)$ with different values of $a$ is given in figure 7. From the plot, it is clear that $g_0(x) = x$ has a real solution for all $a \geq 0$. On the other hand, for the region $a \leq 0$ there exist a critical value $\tilde{a}_c(\beta, \mu)$ such that for $a < a_c$ there are no real solutions anymore. Notice that the critical value for $a$ is in the regime $a \leq 0$ when studying thermodynamics, which is different from the critical value for the finite volume spectrum. This is again the same as the relativistic case where one sign of the deformation parameter leads to complex spectrum for high energy states while the other sign leads to the Hagedorn behavior of the partition function.

The qualitative feature for the $T\overline{T}$ deformed theory is the same. We will see that the effect for the deformation is a shift of the chemical potential. This shift can be determined by the self-consistency relations like (179). For $\lambda \leq 0$, the solution always exist while for $\lambda > 0$, there is a critical value $\tilde{\lambda}_c(\beta, \mu)$ beyond which the system breaks down.

We can also interpret this result in a different way in order to make contact to the thermodynamics in the relativistic case. It is known that for fixed $\lambda$, the partition function exhibit a Hagedorn behavior. This means there is an upper bound on the temperature. At the current situation, if we fix $\mu$ and $\lambda$, then there exists a critical value of $\beta_c$ beyond which the self-consistency relation does not have real solution, which signifies a singularity of the system. This can be seen from figure 8.

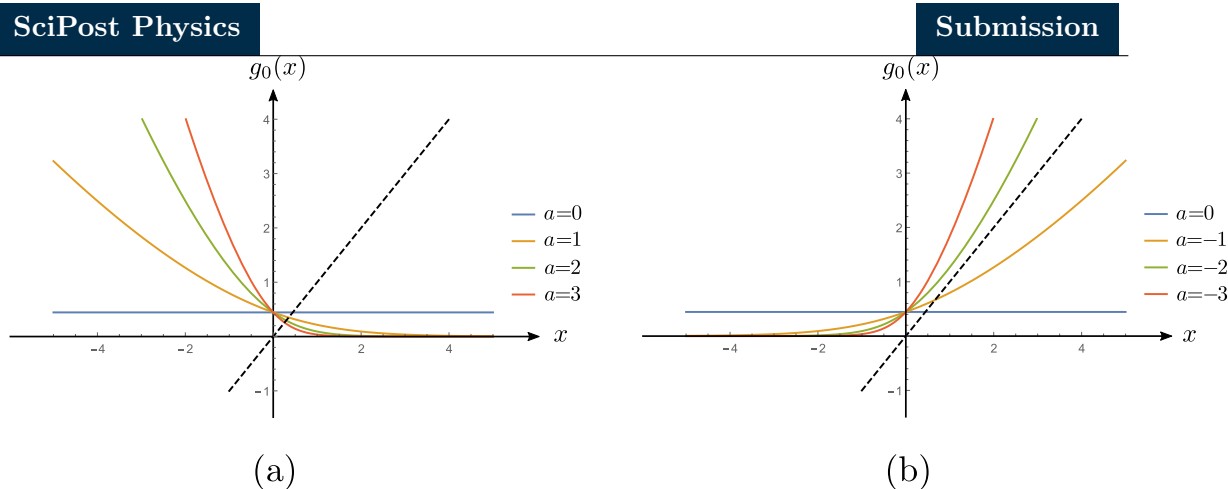

(a) (b)

Figure 7: Plot of $g_0(x)$ for $\beta = 1$, $\mu = 1$ and different values of $a$. In the left panel, we take $a \geq 0$ and in the right panel we take $a \leq 0$. The dashed line is the plot for the function $f(x) = x$. We see when $a \geq 0$, the solution $x = g_0(x)$ always has a real solution. For $a \leq 0$, there is a critical value $a_c$ such that for $a < a_c$, there is no real solution for the equation $x = g_0(x)$.

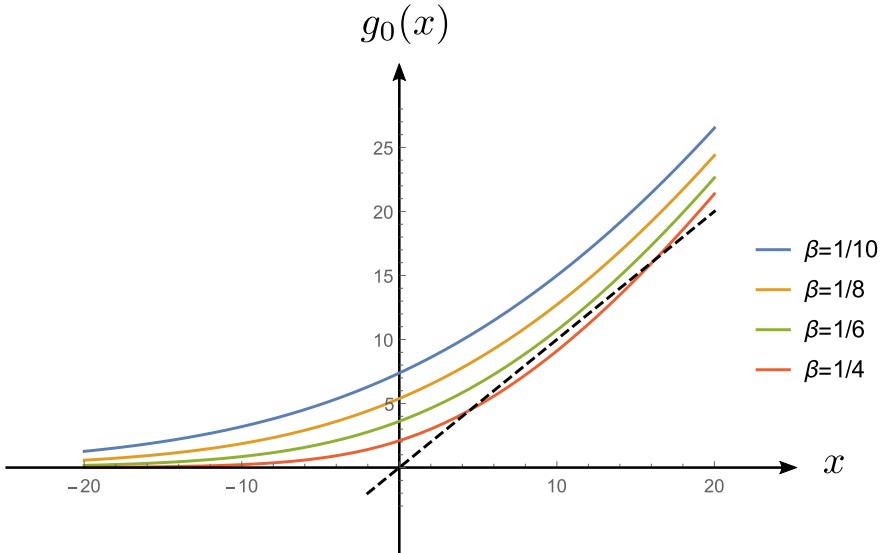

Figure 8: Plot of $g_0(x)$ for fixed $a = 1$, $\mu = 1$ and different values of $\beta$. The dashed line is the plot for the function $f(x) = x$. We see that there's a lower bound of $\beta_c$ below which the system breaks down. This implies that there's a upper bound on the temperature, which is the Hagedorn temperature.

This is the non-relativistic counterpart of the Hagedorn behavior. It is further argued in [22] that the singularity is a branch point.

The physical interpretation for the Hagedorn like behavior is as follow. For $a < 0$, the separations between the particles become larger, which decreases the difference between energy levels. In order words, the energy levels become more dense and the density of states $\rho(E)$ grows faster.

---

[5]In this section, we consider the hard rod fermionic model. This is slightly different from the bosonic model which was considered in section 2.

This in turn increases the entropy and lead to the singularity in the partition function.

## 8.2 The free fermion limit

Now we consider the $T\overline{T}$ deformation of the Lieb-Liniger model in the free fermion limit $c \to \infty$. The deformed TBA equation reads

$$\varepsilon_\lambda(u) = u^2 - \mu + \frac{\lambda}{2\pi\beta} \int_{-\infty}^{\infty} \left(2uv - v^2\right) \ln\left(1 + e^{-\beta\varepsilon_\lambda(v)}\right) \mathrm{d}v \tag{181}$$

This integral equation can be solved using the method in appendix C. The solution takes the following form

$$\varepsilon_\lambda(u) = u^2 - \mu + \lambda(2u\,G_1 - G_2) \tag{182}$$

where $G_1$ and $G_2$ are the solutions of the following self-consistency equations

$$G_k = \frac{1}{2\pi\beta} \int_{-\infty}^{\infty} v^k \ln\left(1 + e^{-\beta(v^2 - \mu + 2\lambda G_1 v - \lambda G_2)}\right) \mathrm{d}v, \qquad k = 1, 2. \tag{183}$$

Or equivalently,

$$G_1 = \frac{\lambda G_1}{2\sqrt{\pi}\beta^{3/2}} \mathcal{F}_{\frac{1}{2}}(\eta), \tag{184}$$

$$G_2 = \frac{\lambda^2 G_1^2}{2\sqrt{\pi}\beta^{3/2}} \mathcal{F}_{\frac{1}{2}}(\eta) + \frac{1}{4\sqrt{\pi}\beta^{5/2}} \mathcal{F}_{\frac{3}{2}}(\eta)$$

with

$$\eta = \beta(\mu + \lambda G_2 + \lambda^2 G_1^2). \tag{185}$$

The deformed pressure is given by

$$\mathrm{P}_\lambda = \frac{1}{2\pi\beta} \int_{-\infty}^{\infty} \ln\left(1 + e^{-\beta\varepsilon_\lambda(u)}\right) \mathrm{d}u = \frac{1}{2\sqrt{\pi}\beta^{3/2}} \mathcal{F}_{\frac{1}{2}}(\eta). \tag{186}$$

The first equation of (184) can be solved by $G_1 = 0$ or

$$1 - \frac{\lambda}{2\sqrt{\pi}\beta^{3/2}} \mathcal{F}_{\frac{1}{2}}(\eta) = 0 \tag{187}$$

If we take the solution (187) and plug in (186), we find $\mathrm{P}_\lambda = 1/\lambda$, which is divergent at $\lambda = 0$. This is non-physical as we expect in the $\lambda \to 0$ limit we should recover the undeformed result. Therefore we conclude that we should take $G_1 = 0$. The equation for $G_2$ simplifies to

$$G_2 = \frac{1}{4\sqrt{\pi}\beta^{5/2}} \mathcal{F}_{\frac{3}{2}}(\beta(\mu + \lambda G_2)) \tag{188}$$

This self-consistency equation can be compared to (179). At small $\lambda$, we can solve the equation perturbatively

$$G_2 = \frac{1}{4\sqrt{\pi}\beta^{5/2}} F_{3/2} + \frac{\lambda}{16\pi\beta^4} F_{3/2}F_{1/2} + \frac{\lambda^2}{128\pi^{3/2}\beta^{11/2}} F_{3/2}(2F_{1/2}^2 + F_{3/2}F_{-1/2}) + \cdots \tag{189}$$

where

$$F_k = -\mathrm{Li}_{k+1}\left(-e^{\beta\mu}\right) = -\mathrm{Li}_{k+1}\left(-z\right) \tag{190}$$

For finite value of $\beta, \mu, \lambda$, the value of $G_2$ can be found numerically. Similar to the hard rod case, for $\lambda < 0$, we can always find a real solution for $G_2$. For $\lambda > 0$, there is a critical value $\tilde{\lambda}_c(\beta, \mu)$. Or equivalently, for fixed $\lambda > 0$ and $\mu$, there is a Hagedorn temperature $1/\beta_H(\mu, \lambda)$ beyond which the system breaks down.

To gain a more analytical expression for the Hagedorn temperature, we can consider the classical limit. The self-consistency relation (205) can be written equivalently as (after an integration by part)

$$G_2 = \frac{1}{3\pi} \int_{-\infty}^{\infty} \frac{u^4}{1 + e^{\beta(u^2 - \mu - \lambda G_2)}} \mathrm{d}u \tag{191}$$

In the high temperature or low density limit, we can approximate the Fermi-Dirac distribution by the classical Maxwell-Boltzmann distribution, which leads to

$$G_2 = \frac{2}{3\pi} \int_{-\infty}^{\infty} u^4 e^{-\beta(u^2 - \mu - \lambda G_2)} \mathrm{d}u \tag{192}$$

where $G_2$ can be factorized out from the integral. Defining $W = -\beta\lambda G_2$, the self-consistency relation can be brought to the form

$$We^W = z \tag{193}$$

where

$$z = -\frac{2\beta\lambda}{3\pi} \int_{-\infty}^{\infty} u^4 e^{-\beta(u^2 - \mu)} \mathrm{d}u = -\frac{e^{\beta\mu}\lambda}{2\sqrt{\pi}\beta^{3/2}} \tag{194}$$

The equation (193) can be solved by the Lambert $W$-function $W = W_0(z)$ where $W_0$ is the principal branch. It is well-known that (193) only has real solutions for $z \geq -e^{-1}$, which leads to

$$\frac{e^{\beta\mu}\lambda}{2\sqrt{\pi}\beta^{3/2}} \leq \frac{1}{e}. \tag{195}$$

This condition is always satisfied for $\lambda \leq 0$, which is consistent with our numerical analysis. For fixed $\lambda > 0$, the critical value is given by $\lambda_c(\beta, \mu) = 2\sqrt{\pi}\beta^{3/2}e^{-\beta\mu - 1}$.

To summarize, the effect of T$\overline{\mathrm{T}}$ deformation is shifting the chemical potential

$$\varepsilon(u) = u^2 - \mu(\lambda), \qquad \mu(\lambda) = \mu + \lambda\, G_2(\beta, \lambda) \tag{196}$$

and the amount of shift can be determined from the self-consistency relation within the range where the system is well behaved. This gives the deformed pseudo-energy, from which all the thermodynamics quantities follow. For example, the pressure is given by

$$\mathrm{P}_\lambda = \frac{1}{2\pi\beta} \int_{-\infty}^{\infty} \ln\left(1 + e^{-\beta(u^2 - \mu(\lambda))}\right) \mathrm{d}u \tag{197}$$

## 8.3 The free boson limit

Now we consider the free boson limit. For $c \to 0$, we have $\varphi(u,v) \to 2\pi\delta(u-v)$. The TBA equation simplifies to

$$\varepsilon(u) = u^2 - \mu - \frac{1}{\beta} \ln\left(1 + e^{-\varepsilon(u)/T}\right) \tag{198}$$

which can be solved

$$\varepsilon(u) = \frac{1}{\beta} \ln\left(e^{\beta(u^2-\mu)} - 1\right) \tag{199}$$

The pressure in this case is given by

$$\mathrm{P} = -\frac{1}{2\pi\beta} \int_{-\infty}^{\infty} \ln\left(1 - e^{-\beta(u^2-\mu)}\right) \mathrm{d}u. \tag{200}$$

The T$\overline{\text{T}}$ deformed equation becomes

$$\varepsilon(u) = u^2 - \mu - \frac{1}{\beta} \ln\left(1 + e^{-\beta\varepsilon(u)}\right) + \frac{\lambda}{2\pi\beta} \int_{-\infty}^{\infty} (2uv - v^2) \ln\left(1 + e^{-\beta\varepsilon(v)}\right) \mathrm{d}v \tag{201}$$

Inspired by the fermionic case, we search for the solution of the following form

$$\varepsilon_\lambda(u) = \frac{1}{\beta} \ln\left(\exp\left(\beta[u^2 + 2u\lambda B_1 - (\mu + \lambda B_2)]\right) - 1\right) \tag{202}$$

Plugging into (201), we find that $B_1$ and $B_2$ satisfies the following self-consistency relations

$$B_k = \frac{1}{2\pi\beta} \int_{-\infty}^{\infty} u^k \ln\left(1 - e^{-\beta(u^2 + 2u\lambda B_1 - \mu - \lambda B_2)}\right) \mathrm{d}u, \qquad k = 1, 2. \tag{203}$$

From a similar analysis to the fermionic case, we find that $B_1 = 0$ and we have only one self-consistency relation

$$B_2 = \frac{1}{2\pi\beta} \int_{-\infty}^{\infty} u^2 \ln\left(1 - e^{-\beta(u^2-\mu-\lambda B_2)}\right) \mathrm{d}u \tag{204}$$

Or equivalently

$$B_2 = -\frac{1}{4\sqrt{\pi}\beta^{5/2}} \mathcal{B}_{\frac{3}{2}}\left(\beta(\mu + \lambda B_2)\right) \tag{205}$$

where $\mathcal{B}_s(\eta)$ is the Bose-Einstein integral (see appendix D). Therefore, we find once again that the effect of the T$\overline{\text{T}}$ deformation is shifting the chemical potential and the shifted amount is given by the solution of the self-consistency relation. The qualitative features is the same as the free fermion case and there is an upper bound on the temperature.

## 8.4 Generic coupling

From the result of the previous cases, it is now clear what we shall expect at finite $c$. For generic coupling $c$, we have the TBA equation

$$\varepsilon_\lambda(u,\mu) = u^2 - \mu - \frac{1}{2\pi\beta} \int_{-\infty}^{\infty} \varphi(u,v) \ln\left(1 + e^{-\beta\varepsilon_\lambda(u,\mu)}\right) \tag{206}$$

$$+ \frac{\lambda}{2\pi\beta} \int (2uv - v^2) \ln\left(1 + e^{-\beta\varepsilon_\lambda(u,\mu)}\right)$$

where we have written the explicit dependence of chemical potential $\mu$ in the pseudo-energy. From the experience of the free cases, we look for a solution of the following form

$$\varepsilon_\lambda(u, \mu) = \varepsilon_0 \left( u + \lambda A_1, \mu + \lambda A_2 + \lambda^2 A_1^2 \right) \tag{207}$$

where $\varepsilon_0(u, \mu)$ is the undeformed pseudo-energy which we assume to be a known function. Plugging this into the original equation, we find the self-consistency relation for $A_k$

$$A_k = \frac{1}{2\pi\beta} \int_{-\infty}^{\infty} v^k \ln \left( 1 + e^{-\varepsilon_0(u+\lambda A_1, \mu+\lambda A_2+\lambda^2 A_1^2)} \right) \mathrm{d}v \tag{208}$$

Before the deformation, the pseudo-energy $\varepsilon(u, \mu)$ is an even function of $u$. From the analysis of the free theories, we know that after the deformation $\varepsilon_\lambda(u, \mu)$ is still an even function of $u$. Physically it is a reasonable expectation that for generic $c$ the deformed pseudo-energy is still even like in the free cases, which allows us to set $A_1 = 0$. The self-consistency relation simplifies to

$$A_2 = \int_{-\infty}^{\infty} v^2 \ln \left( 1 + e^{-\varepsilon_0(u, \mu+\lambda A_2)} \right) \mathrm{d}v \tag{209}$$

The pressure of the system is given by

$$\mathrm{P}_\lambda = \frac{1}{2\pi\beta} \int_{-\infty}^{\infty} \ln \left( 1 + e^{-\varepsilon_0(u, \mu+\lambda A_2)} \right) \tag{210}$$

To compute the deformed quantity, we need to find the solution of (209), which can be done either perturbatively or numerically. The qualitative feature is the same as the free cases and we do not repeat here.

# 9 GGE and higher bilinear deformations

In this section, we make some comments about integrable bilinear deformations and the generalized Gibbs ensemble. This is useful for the study of out-of-equilibrium physics of Lieb-Liniger model. We will show that the shift in chemical potential and the self-consistency relation is a general feature for the bilinear deformation. A similar analysis for relativistic integrable QFTs has been done recently in [31].

We consider a generalized Hamiltonian and density matrix

$$H(\{\beta\}) = \sum_{n=0}^{\infty} \beta_n Q_n, \qquad \hat{\rho}_{\mathrm{GGE}} = \exp \left( -\sum_{n=0}^{\infty} \beta_n Q_n \right) \tag{211}$$

where $\{\beta\}$ is a set of generalized chemical potential. Since the eigenstate $|\mathbf{u}_N\rangle$ diagonalize all the charges simultaneously, it also diagonalizes the generalized Hamiltonian

$$H(\{\beta\})|\mathbf{u}_N\rangle = E_N(\{\beta\}|\mathbf{u})|\mathbf{u}_N\rangle \tag{212}$$

where

$$E_N(\{\beta\}|\mathbf{u}) = \sum_{j=1}^{N} e_0(\{\beta\}|u_j), \qquad e_0(\{\beta\}|u) = \sum_{n} \beta_n h_n(u). \tag{213}$$

Following the standard procedure, one can derived the generalized TBA equation

$$\epsilon(u) = e_0(\{\beta\}|u) - \frac{1}{2\pi} \int_{-\infty}^{\infty} \varphi(u,v) \ln\left(1 + e^{-\epsilon(v)}\right) dv. \tag{214}$$

By taking $\epsilon(u) = \varepsilon(u)/\beta$ and $\beta_0 = -\mu, \beta_2 = \beta$ and the rest $\beta_n = 0$ in $e_0(u)$, we recover the usual TBA which we consider in the previous sections. Now we turn on the bilinear deformation triggered by $O_{a,b}$. This changes the TBA kernel in (214) as

Specifying to Lieb-Liniger model, we have $h_a'(u) = au^{a-1} = ah_{a-1}(u)$. Therefore we can rewrite the TBA equation as

$$\epsilon(u) = e_0(\{\beta\}|u) - \frac{1}{2\pi} \int_{-\infty}^{\infty} \varphi_\lambda(u,v) \ln\left(1 + e^{-\epsilon(v)}\right) dv \tag{216}$$

$$= e_0(\{\tilde{\beta}\}|u) - \frac{1}{2\pi} \int_{-\infty}^{\infty} \varphi(u,v) \ln\left(1 + e^{-\epsilon(v)}\right) dv$$

which takes the same form as the original TBA with shifted generalized chemical potential. More explicitly, the deformed chemical potentials are

$$\beta_{a-1} \mapsto \beta_{a-1} + \lambda a\, G_b, \qquad \beta_{b-1} \mapsto \beta_{b-1} - \lambda b\, G_a \tag{217}$$

while the other chemical potentials are unaffected. The shifts $G_a$ and $G_b$ can be determined by the similar self-consistency relation as we discussed before. Let us denote the undeformed generalized pseudo-energy as $\epsilon_0(\{\beta\}|u)$. The self consistency relation reads

$$G_k = \int_{-\infty}^{\infty} h_k(u) \ln\left(1 + e^{-\epsilon_0(\{\tilde{\beta}\}|u)}\right) du. \tag{218}$$

We expect that such consistency relations have similar behavior to the ones we discussed before, which can be seen by analyzing them in certain limits such as the free fermion and free boson limits.

The deformed spectrum is well-defined for $\lambda > 0$, but exhibit a critical value $\lambda_c$ for $\lambda < 0$ for fixed particle number $N$ and volume of the system $R$. Alternatively, for fixed $\lambda < 0$ and $N$, there is a lower bound for the system size $R_c$ such that the theory is only well-defined for $R > R_c$.

We use TBA to study thermodynamics of the system. It is found that $T\overline{T}$ deformation changes the chemical potential at finite temperature. The change in the chemical potential is determined by the self-consistency condition (209). This equation has real solutions for all $\lambda < 0$. For $\lambda > 0$, real solution only exist for $0 < \lambda \leq \lambda_c$ for certain critical value of $\lambda_c$, with depends on the temperature $1/\beta$ and the undeformed chemical potential $\mu$. Alternatively, for fixed $\lambda$ and $\mu$, we obtain an upper bound for the temperature, which can be seen as the non-relativistic Hagedorn temperature.

There are many possible future directions that one can pursue in the near future. Spectrum and thermodynamics only captures part of the interesting physics of the deformed model. There are other quantities that we would like to study further. One of the most interesting quantities are the correlation functions. This quantity turns out to be much harder to study in QFT although important progress have been made. By far, we do not have an explicit expression for correlation functions that are *non-perturbative* in $\lambda$ in QFT. We believe Lieb-Liniger model is simpler than QFT and hopefully we could make more progress in this model, which may shed new lights on correlation functions in other theories.

In this paper we mainly focus on the repulsive case of the Lieb-Liniger model where $c > 0$. The attractive regime $c < 0$ is also interesting. In this case, we have bound states and it is interesting to see how this fact modifies various quantities in the deformed theory. Also, this model is related to other integrable models such as supersymmetric field theories [32] and random matrix model [33]. This might leads to natural definitions for new kinds of solvable deformations for these models.

$T\overline{T}$ deformation for QFT can be interpreted as coupling the theory to a 2d topological gravity. The fact that the $T\overline{T}$ changes the size of the system strongly suggests that there should be some relation between the deformation and coupling the theory to certain kind of non-relativistic gravity theory. This can be most naturally done in the framework of Newton-Cartan theory.

Finally it is also interesting to study out-of-equilibrium physics of the deformed theory. For integrable models, this can be done by the powerful method of generalized hydrodynamics (see the lecture note [34] for a nice introduction). A study of such kind has been perform for CFTs in [35] using both GHD and holography. We expect some of the main features should also be present in our case.

### Acknowledgements

I thank Balazs Pozsgay and Gabor Takacs for collaborations on related works. I also thank Benjamin Doyon and John Cardy for helpful correspondences and Shouvik Datta for discussions. I thank Lily Jiang for inspiration.

## A    Inviscid Burgers' equation

In this appendix, we briefly discuss some properties of the inviscid Burgers' equation that is useful in the main text. The inviscid Burgers' equation reads

$$\frac{\partial u}{\partial t} + u\frac{\partial u}{\partial x} = 0 \tag{219}$$

where $u(x,t)$ is a function of the space $x$ and time $t$. Comparing with the flow equation of the energy in the zero momentum sector, we find that we can identify $R$ with $x$, $-\lambda$ with $t$ [6]. Below we will follow the notation in the main text.

---

[6]In Burgers' equation the shock is formed at certain $t > 0$. That's why the singularity in the spectrum occur at $\lambda < 0$.

**Method of characteristic** The Burgers' equation can be solved by method of characteristics. The characteristic refers to a trajectory $R(\lambda)$ on the $(R, \lambda)$ plane which satisfies the following equation

$$\frac{\mathrm{d}}{\mathrm{d}\lambda} R(\lambda) = -E(R(\lambda), \lambda). \tag{220}$$

Then from the Burgers' equation, it is easy to see that on each characteristics $E(R(\lambda), \lambda)$ is constant

$$\frac{\mathrm{d}}{\mathrm{d}\lambda} E(R(\lambda), \lambda) = \partial_\lambda E(R(\lambda), \lambda) + \partial_R E(R(\lambda), \lambda) \, R'(\lambda) \tag{221}$$

$$= \partial_\lambda E(R(\lambda), \lambda) - E(R(\lambda), \lambda) \partial_R E(R(\lambda), \lambda) = 0$$

where we have used the Burgers' equation in the second line. Since it is a constant, we can evaluate $E(R(\lambda), \lambda)$ at any point of $\lambda$. In particular, we can take $\lambda = 0$ and denote $\xi = R(0)$. Then we can write $E(R(\lambda), \lambda) = E(\xi, 0)$. Then for each point $(R, t)$, we can solve

$$R = \xi - E(\xi, 0)\lambda. \tag{222}$$

for $\xi$ and we have

$$E(R, \lambda) = E(\xi, 0). \tag{223}$$

Therefore, if we are given the initial profile $E(x, 0)$ as a function of $x$. To determine the value of $E(R, t)$ at any point $(R, \lambda)$, we first solve the equation (222) to find $\xi(R, \lambda)$, and then plug the solution on the right hand side of (223).

**Shock formation** The method described above can be used when $\lambda$ is small. On the other hand, for large enough $\lambda$ it can happen that the solution of (222) is not unique. The reason is that several characteristic may cross each other. This will eventually happen whenever the initial profile $\partial_R E(R, 0)$ is negative at any point. Suppose at some $\lambda_c < 0$ some characteristics first cross. At this point, the $E(R, \lambda)$ has an infinite slope, namely $\partial_R E(R, \lambda)$ is divergent. We say that the wave breaks and a shock forms. This is precisely the point where the deformed spectrum becomes complex.

The shock formation is characterized by the following equation

$$1 - t\partial_\xi E(\xi, 0) = 0. \tag{224}$$

Let us denote the solution by $\xi_c$. At this value, the radius $R_c$ is given by

$$R_c = \xi_c - tE(\xi_c, 0). \tag{225}$$

The physical interpretation in our case is that, for fixed value of $\lambda$, the radius cannot be smaller than $R_c$. Namely, beyond that UV scale, the theory breaks down.

As an example, let us consider the free fermion limit of Lieb-Liniger model in the main text. The initial profile is given by $E(R, 0) = \alpha_N/R^2$. According to (224), shock formation occur at

$$1 + \frac{2\lambda\alpha_N}{\xi_c^3} = 0, \qquad \xi_c^3 = -2\alpha_N\lambda. \tag{226}$$

and the critical value of $R_c$ is given by

$$R_c = \frac{1}{\xi_c^2}(\xi_c^3 - \lambda\alpha_N) = -\frac{3\alpha_N\lambda}{\xi_c^2} \tag{227}$$

which leads to the following result

$$4R_c^3 = -27\alpha_N\lambda. \tag{228}$$

Equivalently this equation can be interpreted as if we fix $R$ and $\alpha_N$, the critical value of $\lambda_c$ is given by $\lambda_c = -4R^3/(27\alpha_N)$, which is the same as what we found in the main text (100). For more complicated initial profile $E(R,0)$, we can solve (224) and (225) numerically to find the critical value of $R_c$ for fixed $\lambda$, or find the critical value of $\lambda_c$ of fixed $R$.

# B    Bilocal operator and scattering states

In this appendix, we show that the partially ordered state is an eigenvector of the bilocal operator

$$X_{\mathcal{J}\mathcal{J}} = \int_{x<y} \mathrm{d}x\mathrm{d}y \, q_1(x)q_2(y). \tag{229}$$

Let us denote the partially ordered state such that $|u_1 < u_2 < \cdots < u_N\rangle$. This states consists $N$-particles with rapidities $u_1, u_2, \cdots, u_N$ at positions $x_1, x_2, \cdots, x_N$ such that $x_1 < x_2 < \cdots < x_N$. We will prove that

$$X_{\mathcal{J}\mathcal{J}}|u_1 < u_2 < \cdots < u_N\rangle = \left(\sum_{i=1}^{N} f(u_i) + \sum_{i<j}^{N} h_1(u_i)h_2(u_j)\right)|u_1 < u_2 < \cdots < u_N\rangle. \tag{230}$$

where $h_1(u)$ and $h_2(u)$ are the eigenvalues of one-particle state, which measures the charge of the particle with rapidity $u$, and $f(u_i)$ denotes the eigenvalue of both operators acting on the same particle with rapidity $u_i$. We first prove this explicitly for the case $N = 3$, the generalization for higher particles is straightforward.

Consider a 3-particle state $|u_1 < u_2 < u_3\rangle$. The action of $X_{\mathcal{J}\mathcal{J}}$ on the state is given by

$$\int_{x<y} \mathrm{d}x\mathrm{d}y \, q_1(x)q_2(y)|u_1 < u_2 < u_3\rangle \tag{231}$$

If the integrals of $q_1(x), q_2(y)$ sweep over some particles, we collect the charges of these particles. The integration domain is given by the shaded parts in figure 9. We compute the integral by decomposing the integral domain into 5 disconnected parts, labeled by I,II,III,IV,V respectively. In the leftmost region-I, the integral over $q_1(x)$ does not contain any particles, so the result is vanishing. Likewise, the integral over region-V is vanishing. Now consider region-II, the $q_1(x)$ integral contains the particle $u_1$ and the $q_2(y)$ integral contains $u_1, u_2, u_3$. When both $q_1(x)$ and $q_2(y)$ act on the same particle $u_1$, the result is denoted by $f(u_1)$. So the integral over region-II leads to

$$\text{region-II} = f(u_1) + h_1(u_1)[h_2(u_2) + h_2(u_3)] \tag{232}$$

Similarly, we can see that the integral over region-III,IV are given by

$$\text{region-III} = f(u_2) + h_1(u_2)h_2(u_3), \qquad \text{region-IV} = f(u_3). \tag{233}$$

Summing over the contributions, we find

$$f(u_1) + f(u_2) + f(u_3) + h_1(u_1)[h_2(u_2) + h_2(u_3)] + h_1(u_2)h_2(u_3). \tag{234}$$

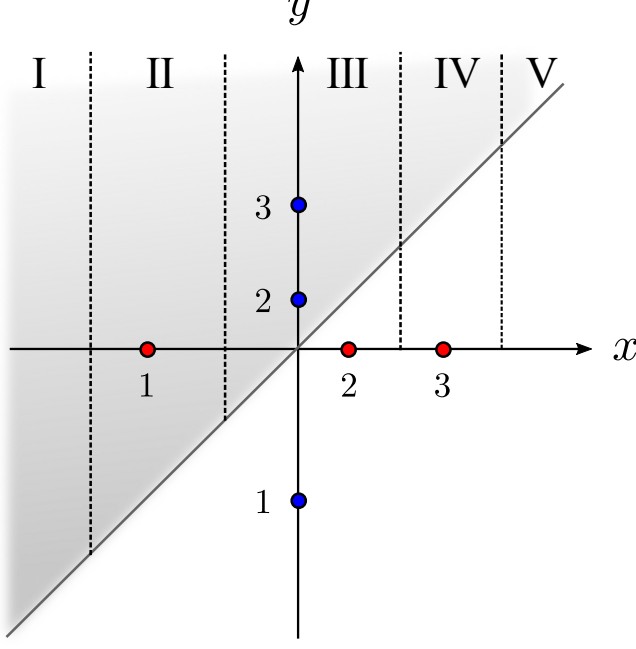

Figure 9: Integration domain of the bilocal operator. The shaded part is the domain for $x < y$.

It is straightforward to generalize the above argument to $N$-particle state. The eigenvalue is given by

$$\sum_{i=1}^{N} f(u_i) + \sum_{i<j}^{N} h_1(u_i)h_2(u_j). \tag{235}$$

## C    Integral equations

In this appendix, we discuss the solution of certain integral equations with degenerate kernels.

### C.1    Fredholm equation

The Fredholm equation of the second kind refers to the following type of integral equation

$$\rho(u) = f(u) + \lambda \int_a^b K(u,v)\rho(v)\mathrm{d}v \tag{236}$$

A kernel is called *degernate* if it takes the following form

$$K(u,v) = \sum_{k=1}^{n} g_k(u)h_k(v) \tag{237}$$

The equation can be written as

$$\rho(u) = f(u) + \lambda \sum_{k=1}^{n} g_k(u) \int_a^b h_k(v)\rho(v)\mathrm{d}v \tag{238}$$

Assuming the equation has a solution, we introduce the notation

$$A_k = \int_a^b h_k(v)\rho(v)\mathrm{d}v \tag{239}$$

Then the equation is given by

$$\rho(u) = f(u) + \lambda \sum_{k=1}^{n} A_k g_k(u) \tag{240}$$

Now multiply both sides by $h_m(u)$ and integrate from $a$ to $b$, we obtain

$$A_m = f_m + \lambda \sum_{k,m=1}^{n} s_{m,k} A_k \tag{241}$$

where

$$s_{m,k} = \int_a^b h_m(u)g_k(u)\mathrm{d}u, \qquad f_m = \int_a^b h_m(u)f(u)\mathrm{d}u \tag{242}$$

which are known functions. Solving the algebraic equation (241) gives $A_k$. Using the equation (240), we find the solution $\rho(u)$.

Now we specify to our case. The kernel under consideration is

$$K(u,v) = -(e'(u)p(v) - p'(u)e(v)) \tag{243}$$

where $e(u) = u^2$ and $p(u) = u$ are the energy and momentum of a single excitation. We have

$$\rho(u) = \frac{1}{2\pi} - \frac{\lambda}{2\pi}\left[\mathbb{P}\, e'(u) - \mathbb{E}\, p'(u)\right] \tag{244}$$

Multiplying both sides by $p(u)$ and $e(u)$ and integrate, we find that

$$\mathbb{P} = \frac{1}{2\pi}f_1 - \frac{\lambda}{2\pi}\left[s_{12}\mathbb{P} - s_{11}\,\mathbb{E}\right], \tag{245}$$

$$\mathbb{E} = \frac{1}{2\pi}f_2 - \frac{\lambda}{2\pi}\left[s_{22}\mathbb{P} - s_{21}\,\mathbb{E}\right]$$

where

$$f_1 = \int_a^b p(u)\mathrm{d}u, \qquad f_2 = \int_a^b e(u)\mathrm{d}u. \tag{246}$$

$$s_{11} = \int_a^b p(u)p'(u)\mathrm{d}u, \qquad s_{12} = \int_a^b p(u)e'(u)\mathrm{d}u, \tag{247}$$

$$s_{22} = \int_a^b e(u)e'(u)\mathrm{d}u, \qquad s_{21} = \int_a^b e(u)p'(u)\mathrm{d}u$$

Solving these equations, we find that

$$\mathbb{P} = \frac{(2\pi - \lambda s_{21}) + \lambda s_{11}f_2}{4\pi^2 + 2\pi(s_{12} - s_{21})\lambda + (s_{11}s_{22} - s_{12}s_{21})\lambda^2}, \tag{248}$$

$$\mathbb{E} = \frac{(2\pi + \lambda s_{12})f_2 - \lambda s_{22}f_1}{4\pi^2 + 2\pi(s_{12} - s_{21})\lambda + (s_{11}s_{22} - s_{12}s_{21})\lambda^2}$$

The deformed density is then given by (244). Plugging in the explicit dispersion relations $p(u) = u$ and $e(u) = u^2$, we find more explicit expressions. Let us introduce the following notation

$$M_k \equiv \int_a^b u^k \mathrm{d}u. \tag{249}$$

Then we have

$$f_1 = M_1, \quad f_2 = M_2, \quad s_{11} = M_1, \quad s_{12} = 2M_2, \quad s_{21} = M_2, \quad s_{22} = 2M_3. \tag{250}$$

## C.2   Urysohn equation

The TBA equation in the Girardeau-Tonks limit takes form of the Urysohn equation of the second kind with a degenerate kernel. We discuss the solution of this type of equation. Consider the following equation

$$y(u) + \sum_{k=1}^n \int_a^b g_k(u) F_k(v, y(v)) \mathrm{d}v = f(u) \tag{251}$$

This equation has the solution of the form

$$y(u) = f(u) + \sum_{k=1}^n A_k \, g_k(u) \tag{252}$$

where $A_k$ are the solution of the following system of equations (which can be algebraic or transcendental)

$$A_m + \int_a^b F_m \left( u, f(u) + \sum_{k=1}^n A_k \, g_k(u) \right) \mathrm{d}u = 0 \tag{253}$$

# D   Some integrals

In this appendix, we give some useful formula for the integrals of the form

$$\int_{-\infty}^\infty x^n \ln \left( 1 \pm e^{-ax^2 + bx + c} \right) \mathrm{d}x, \qquad \mathrm{Re}[a] > 0, \quad n \in \mathbb{N} \tag{254}$$

They appear in the computation of deformed TBA in the main text. In particular, we are interested in the cases for $n = 0, 1, 2$. To compute these integrals, we first rewrite the function $\ln(1 \pm e^{-X})$ as an infinite series and then perform the integral for each term using the following formulae

$$\int_{-\infty}^\infty e^{-ax^2 + bx + c} \mathrm{d}x = \frac{\sqrt{\pi}}{\sqrt{a}} e^{\frac{b^2}{4a} + c}, \tag{255}$$

$$\int_{-\infty}^\infty x \, e^{-ax^2 + bx + c} \mathrm{d}x = \frac{\sqrt{\pi} b}{2a^{3/2}} e^{\frac{b^2}{4a} + c},$$

$$\int_{-\infty}^\infty x^2 \, e^{-ax^2 + bx + c} \mathrm{d}x = \frac{\sqrt{\pi}(b^2 + 2a)}{4a^{5/2}} e^{\frac{b^2}{4a} + c}.$$

The integrals can be written in terms of *Fermi-Dirac integrals* $\mathcal{F}_s(\eta)$ and *Bose-Einstein integrals* $\mathcal{B}_s(\eta)$ which are related to the polylogarithm as

$$\mathcal{F}_s(\eta) = -\text{Li}_{s+1}(-e^\eta), \qquad \mathcal{B}_s(\eta) = \text{Li}_{s+1}(e^\eta) \tag{256}$$

After taking the infinite sum, we find

$$\int_{-\infty}^{\infty} \ln\left(1 + e^{-ax^2+bx+c}\right) dx = \frac{\sqrt{\pi}}{\sqrt{a}} \mathcal{F}_{\frac{1}{2}}\left(\frac{b^2}{4a} + c\right), \tag{257}$$

$$\int_{-\infty}^{\infty} x \ln\left(1 + e^{-ax^2+bx+c}\right) dx = \frac{\sqrt{\pi}b}{2a^{3/2}} \mathcal{F}_{\frac{1}{2}}\left(\frac{b^2}{4a} + c\right),$$

$$\int_{-\infty}^{\infty} x^2 \ln\left(1 + e^{-ax^2+bx+c}\right) dx = \frac{\sqrt{\pi}b^2}{4a^{5/2}} \mathcal{F}_{\frac{1}{2}}\left(\frac{b^2}{4a} + c\right) + \frac{\sqrt{\pi}}{2a^{3/2}} \mathcal{F}_{\frac{3}{2}}\left(\frac{b^2}{4a} + c\right)$$

and

$$\int_{-\infty}^{\infty} \ln\left(1 - e^{-ax^2+bx+c}\right) dx = -\frac{\sqrt{\pi}}{\sqrt{a}} \mathcal{B}_{\frac{1}{2}}\left(\frac{b^2}{4a} + c\right), \tag{258}$$

$$\int_{-\infty}^{\infty} x \ln\left(1 + e^{-ax^2+bx+c}\right) dx = -\frac{\sqrt{\pi}b}{2a^{3/2}} \mathcal{B}_{\frac{1}{2}}\left(\frac{b^2}{4a} + c\right),$$

$$\int_{-\infty}^{\infty} x^2 \ln\left(1 + e^{-ax^2+bx+c}\right) dx = -\frac{\sqrt{\pi}b^2}{4a^{5/2}} \mathcal{B}_{\frac{1}{2}}\left(\frac{b^2}{4a} + c\right) - \frac{\sqrt{\pi}}{2a^{3/2}} \mathcal{B}_{\frac{3}{2}}\left(\frac{b^2}{4a} + c\right)$$

# E Perturbative expansions

In this appendix, we give the energy of the ground state for $N$-particle state in large $c$ expansion. We work out the first two orders explicitly. The BAE we need to solve is

$$u_j R + \frac{2}{c} \sum_{k=1}^{N} (u_j - u_k) - \frac{2}{3c^3} \sum_{k=1}^{N} (u_j - u_k)^3 + \mathcal{O}(c^{-5}) = 2\pi I_j \tag{259}$$

The idea is to find the solution of these equation in the form

$$u_j = u_j^{(0)} + \frac{u_j^{(1)}}{c} + \frac{u_j^{(2)}}{c^2} + \cdots \tag{260}$$

It is easy to find that

$$u_j^{(0)} = \frac{2\pi I_j}{R} \tag{261}$$

The first order equation reads

$$u_j^{(1)} R + 2 \sum_{k=1}^{N} (u_j^{(0)} - u_k^{(0)}) = 0 \tag{262}$$

which can be solved readily

$$u_j^{(1)} = \frac{2\text{M}_1}{R} - \frac{2N}{R} u_j^{(0)} = \frac{2\text{M}_1}{R} - 2n_0 u_j^{(0)} \tag{263}$$

where for later convenience we introduce the following notation

$$\mathrm{M}_k = \sum_{j=1}^{N} \left( \frac{2\pi I_j}{R} \right)^k .$$ (264)

For the ground state, we have $\mathrm{M}_{2n+1} = 0$. The second order equation reads

$$u_j^{(2)} R + 2 \sum_{k=1}^{N} (u_j^{(1)} - u_k^{(1)}) = 0$$ (265)

we find that

$$u_j^{(2)} = \frac{2}{R} \sum_{k=1}^{N} u_k^{(1)} - \frac{2N}{R} u_j^{(1)} = 4n_0^2 u_j^{(0)} - \frac{4n_0 \mathrm{M}_1}{R}$$ (266)

So up to $\mathcal{O}(c^{-2})$, we find that for the ground state Bethe roots

$$u_j = u_j^{(0)} - \gamma u_j^{(0)} + \gamma^2 u_j^{(0)} + \cdots, \qquad \gamma = \frac{2n_0}{c}.$$ (267)

The energy is given by

$$E_N(R, 0; c) = (1 - 2\gamma + \gamma^2) \mathrm{M}_2 + \mathcal{O}(c^{-3})$$ (268)

where for the ground state $\mathrm{M}_2 = \alpha_N / R^2$. Therefore up to $1/c^2$ order, the energy is given by

$$E_N(R, 0; c) = \frac{\alpha_N}{R^2} \left( 1 - \frac{4N}{c} \frac{1}{R} + \frac{4N^2}{c^2} \frac{1}{R^2} \right) + \mathcal{O}(c^{-3})$$ (269)

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
