# Peer review of "$\mathrm{T}\bar{\mathrm{T}}$-deformed 1d Bose gas"

_SciPost Physics_

## Round 1 · Referee Report · Anonymous · 2021-6-8

Strengths

Very active subject matter
Extensive study
Well written
Interesting ideas

Weaknesses

At times repeats or only slightly extends calculations done in other works
Some imprecisions
Perhaps missing a bit of literature

Report

In this paper, the author studies TTbar-type of deformations of non-relativistic integrable models, with the main example being the Lieb-Liniger models, the hard-rod gas and free-Boson and free-Fermion limits. The paper presents an extensive study of the matter, with a good analysis of quantities of physical interest. It has some overlap with paper [22], but the approach and much of the discussion appears to be different, and thus I would say that the two paper complement each other well. The present paper in a sense reproduces many calculations that are already known or simple generalisations of previous works. but it puts together nicely a number of ideas. Overall this is an excellent work and deserves publication, after clarifying aspects of the discussion and derivations as described below.

1)

After eq 27: the term Y_{JJ} can be neglected under the condition that currents vanish. This is nontrivial, as here these are currents associated to generic quantities. For instance, the current of momentum is the pressure, and therefore here it is required that the pressure vanishes at infinity. Thus this is a strong condition on the state in which the system is. This affects the notion of how eigenstates transform. The result is correct in the leading large-volume asymptotic, but the derivation eps 28-32 does not seem to take proper care of this subtlety.

2)

Eq 43 and App B: I am not totally convinced by the derivation of App B leading to eq 43. First, I believe it is not true that sweeping q_i(x) over a particle in the Bethe ansatz wave function leads to a factor of the eigenvalue. If this were the case, the partially-ordered two-particle state would be an eigenstate of the hamiltonian, which it is not. In general, one needs to discard / deal with total derivative terms, leading to boundary contribution within ordered regions, and there the S-matrix plays an important role. Then, likewise it does not appear to be true that there is an eigenvalue f_{12}(u) corresponding to the action of q_1(x) q_2(x) on the same particle. This derivation is an adaptation of the derivation in [7], but the author should clarify the situation here.

3)

Page 12, top: just to note that it is well known that the particle current j_N is equal to the momentum density p in any Galilean model, such as the Lieb-Liniger gas.

4)

Although I leave this to the discretion of the author, I would suggest citing a bit more the literature on GHD, as so many concepts discussed in this work are related to it.

Page 14, top: as the author discusses the average energy current <n|T_{01}|n>, it is perhaps worth mentioning that in integrable models in infinite volume (where an energy eigenstate is a GGE), this was first evaluated in

[a] O. A. Castro-Alvaredo, B. Doyon and T. Yoshimura, Emergent hydrodynamics in integrable quantum systems out of equilibrium, Phys. Rev. X 6, 041065 (2016), arXiv:1605.07331

and

[b] B. Bertini, M. Collura, J. De Nardis, and M. Fagotti, Transport in out-of-equilibrium XXZ chains: exact profiles of charges and currents, Phys. Rev. Lett. 117, 207201 (2016), arXiv:1605.09790

In particular the former does it for the Lieb-Liniger model. See also later works using form factors

[c] Dinh-Long Vu, Takato Yoshimura, Equations of state in generalized hydrodynamics, SciPost Phys. 6, 023 (2019), arXiv:1809.03197

and by other techniques

[d] Herbert Spohn, The collision rate ansatz for the classical Toda lattice, Phys. Rev. E 101, 060103 (2020), arXiv:2004.03802

[e] Takato Yoshimura, Herbert Spohn, Collision rate ansatz for quantum integrable systems, SciPost Phys. 9, 040 (2020), arXiv:2004.07113

The cited papers [29,30] adapt (nontrivially) the formula to finite volumes, presenting a derivation using Bethe ansatz. In fact there is a known general formula for the average current of energy in non-integrable models as well: in non-integrable momentum-preserving models it is found in

[f] Benjamin Doyon, Joseph Durnin, Free energy fluxes and the Kubo-Martin-Schwinger relation, J. Stat. Mech. 2021, 043206 (2021), arXiv:2007.09113

and only depends on the knowledge of the free energy.

Page 25: it should probably be mentioned that generalised current operators were first introduced in

[g] Benjamin Doyon, Takato Yoshimura, A note on generalized hydrodynamics: inhomogeneous fields and other concepts, SciPost Phys. 2, 014 (2017), arXiv:1611.08225

where their infinite-volume averages were written.

Page 26: it is probably worth mentioning that the effective velocity, its involvement in the average currents, and its interpretation as a velocity influenced by a bath of surrounding particle, was discussed in [a,b], and in fact it first appeared in a related context earlier in

[h] L. Bonnes, F. H. L. Essler, A. M. La\"uchli, ’Light-cone’ dynamics after quantum quenches in spin chains, Phys. Rev. Lett. 113, 187203 (2014).

5)

Small things:

The sentence at bottom of page 4 seems to be broken

No period after (9).

After eq 65: these two derivations … do not rely …

Requested changes

Make clarifications of points 1, 2 and 3 of main report
to the discretion of the author, add citations as in point 4
small modifications of point 5.

---

## Round 1 · Referee Report · Anonymous · 2021-6-12

Strengths

1- The paper is well written, with many original comments and results.

2- It is essentially the first paper discussing the TTbar deformation of the Lieb-Liniger model in such details.

3- The background sections are carefully written and very useful.

4-The list of relevant references is essentially complete.

5- The topic is interesting and important.

6- The paper contains many original analytic and numerical results.

Weaknesses

1- Due to the similarity with the relativistic invariant TTbar deformation, some results may look non entirely original.
However, there are also significant differences that make the study of this non-relativistic-invariant model particularly interesting.

2-There is a certain overlap with [22], but this is a longer paper with more details, explanations and results at quantum level and finite temperature/volume.

Report

The author discusses a TTbar-like deformation of the Lieb-Liniger quantum model. The paper is well written, with a detailed description of the background and explanations of all the logical and mathematical steps undertaken. Most of the phenomenology identified resembles that already observed in relativistic-invariant systems, but some crucial differences make this model particularly interesting. I recommend the publication of this paper.

Requested changes

I found some very minor typos:

Above eq. 49 "of the" is repeated twice.

I also recommend a further check of the punctuation at the end of the equations. (See equations (22), (49), (80), (112), (127), (128))

---

## Editorial Decision

editor-in-charge_assigned